



# A review and framework for the evaluation of pixel-level uncertainty estimates in satellite aerosol remote sensing

Andrew M. Sayer[1,2], Yves Govaerts[3], Pekka Kolmonen[4], Antti Lipponen[4], Marta Luffarelli[3], Tero Mielonen[4], Falguni Patadia[1,2], Thomas Popp[5], Adam C. Povey[6], Kerstin Stebel[7], and Marcin L. Witek[8]

[1]GESTAR, Universities Space Research Association, Columbia, MD, USA
[2]NASA Goddard Space Flight Center, Greenbelt, MD, USA
[3]Rayference, Brussels, Belgium
[4]Finnish Meteorological Institute, Atmospheric Research Centre of Eastern Finland, Kuopio, Finland
[5]Deutsches Zentrum für Luft-und Raumfahrt e. V. (DLR), Deutsches Fernerkundungsdatenzentrum (DFD), 82234 Oberpfaffenhofen, Germany
[6]National Centre for Earth Observation, University of Oxford, Oxford, OX1 3PU, UK
[7]NILU - Norwegian Institute for Air Research, NO-2007 Kjeller, Norway
[8]Jet Propulsion Laboratory, California Institute of Technology, 4800 Oak Grove Drive, Pasadena, CA 91109, USA

**Correspondence:** Andrew M. Sayer (andrew.sayer@nasa.gov)

**Abstract.** Recent years have seen the increasing inclusion of per-retrieval prognostic (predictive) uncertainty estimates within satellite aerosol optical depth (AOD) data sets, providing users with quantitative tools to assist in optimal use of these data. Prognostic estimates contrast with diagnostic (i.e. relative to some external truth) ones, which are typically obtained using sensitivity and/or validation analyses. Up to now, however, the quality of these uncertainty estimates has not been routinely

assessed. This study presents a review of existing prognostic and diagnostic approaches for quantifying uncertainty in satellite AOD retrievals, and presents a general framework to evaluate them, based on the expected statistical properties of ensembles of estimated uncertainties and actual retrieval errors. It is hoped that this framework will be adopted as a complement to existing AOD validation exercises; it is not restricted to AOD and can in principle be applied to other quantities for which a reference validation data set is available. This framework is then applied to assess the uncertainties provided by several satellite data sets

(seven over land, five over water), which draw on methods from the empirical to sensitivity analyses to formal error propagation, at 12 Aerosol Robotic Network (AERONET) sites. The AERONET sites are divided into those where it is expected that the techniques will perform well, and those for which some complexity about the site may provide a more severe test. Overall all techniques show some skill in that larger estimated uncertainties are generally associated with larger observed errors, although they are sometimes poorly calibrated (i.e. too small/large in magnitude). No technique uniformly performs best. For

powerful formal uncertainty propagation approaches such as Optimal Estimation the results illustrate some of the difficulties in appropriate population of the covariance matrices required by the technique. When the data sets are confronted by a situation strongly counter to the retrieval forward model (e.g. potential mixed land/water surfaces, or aerosol optical properties outside of the family of assumptions), some algorithms fail to provide a retrieval, while others do but with a quantitatively unreliable uncertainty estimate. The discussion suggests paths forward for refinement of these techniques.





# 1 Introduction

The capability to quantify atmospheric aerosols from spaceborne measurements arguably goes back to 1972 with the launch of the Multispectral Scanner System (MSS) aboard the first Landsat satellite (e.g. Griggs, 1975; Kaufman and Sendra, 1988), primarily designed for land surface characterisation. Earlier satellite-based solar reflectance measurements were (with the exception of the three-colour camera on the Applications Technology Satellite 3, launched 1967) either panchromatic (and used for cloud mapping) or broadband (for radiation). While it was realised from experience with similar sensors on Mars (Hanel et al., 1972) that some aerosols could contribute to signals in the thermal infrared (tIR), they were largely treated as a contaminant in temperature/water vapour retrievals and not routinely quantified (Weaver et al., 2003). Landsat-1 MSS was followed in 1975 by a second Landsat launch and the Stratospheric Aerosol Measurement (SAM) instrument on the Apollo-Soyuz Test Project, a proof-of-concept for monitoring stratospheric aerosols (McCormick et al., 1979), and then by a gradually-expanding variety of instruments from the late 1970s onwards.

At present there are several dozen sensors of various types suitable for quantification of aerosols in flight, and more that have begun and ended operations in between. In addition to the variety of instruments, a variety of algorithms have been developed to retrieve aerosol properties from these measurements (e.g. Kokhanovsky and de Leeuw, 2009; Lenoble et al., 2013; Dubovik et al., 2019, for some reviews of the principles behind various techniques). The majority of these sensors have been used to retrieve total column aerosol optical depth (AOD) across some part(s) of the ultraviolet (UV), visible, near- and shortwave- infrared, and tIR spectral regions, where aerosol particles are optically active; most commonly reported is the mid-visible AOD at a wavelength in the range 500-565 nm. Some sensors are able to retrieve profiles of aerosol extinction, which may be integrated vertically to give partial or total column AOD (dependent on whether or not profiling is possible down to the surface). This proliferation, combined with geophysical and mathematic terminology, makes aerosol remote sensing an incredibly acronym-heavy field; indeed, instruments and algorithms are often referred to by their acronyms rather than full names. Table 1 lists those sensors which have to date been used to process AOD data products, and Table 2 lists those which are able to provide extinction profiles; in many cases, two or more of each type of design, either identical or with small modifications, have been flown. Where multiples of a given sensor have flown the date ranges indicate period(s) of continuous coverage as opposed to launch/decommission dates for individual instruments.

Retrieval algorithms are used to process the calibrated observations (referred to as level 1 or L1 data) to provide level 2 (L2) data products, consisting of geophysical quantities of interest. These L2 products are typically on the L1 satellite observation grid (or a multiple of it) and often further aggregated to level 3 (L3) products on regular space-time grids. For further background and discussion of satellite data processing levels, see Mittaz et al. (2019). Table 3 provides acronyms and full names for some of the L2 processing algorithms which have been applied to L1 measurements from these instruments. Again, many of these algorithms have been applied (identically or with small modification) to multiple sensors. This table is provided as a convenience to the reader to decode acronyms and decrease clutter in later tables and discussion; specific relevant



**Table 1.** Satellite instruments which have been used for column AOD retrieval, arranged by sensor type.

| Acronym | Instrument full name | Orbit(s) | Operation period(s) |
|---|---|---|---|
| | Multispectral imager | | |
| ABI | Advanced Baseline Imager | Geostationary | 2016+ |
| AHI | Advanced Himawari Imager | Geostationary | 2014+ |
| AVHRR | Advanced Very High Resolution Radiometer | Sun-synchronous | 1978+ |
| CAI | Cloud-Aerosol Imager | Sun-synchronous | 2009+ |
| EPIC | Earth Polychromatic Imaging Camera | Lagrange point | 2015+ |
| (E)TM | (Enhanced) Thematic Mapper | Sun-synchronous | 1982+ |
| GOES Imager | Geostationary Operational Environmental Satellite Imager | Geostationary | 1978-2018 |
| GOCI | Geostationary Ocean Color Imager | Geostationary | 2010+ |
| GLI | GLobal Imager | Sun-synchronous | 2002-2003 |
| MERIS | MEdium Resolution Imaging Spectrometer | Sun-synchronous | 2002-2012 |
| MODIS | MODerate resolution Imaging Spectrometer | Sun-synchronous | 2000+ |
| MSS | Multispectral Scanner System | Sun-synchronous | 1972-2013 |
| OLCI | Ocean and Land Color Instrument | Sun-synchronous | 2016+ |
| OLI | Operational Land Imager | Sun-synchronous | 2013+ |
| SeaWiFS | Sea-viewing Wide Field-of-view Sensor | Sun-synchronous | 1997-2010 |
| SEVIRI | Spinning Enhanced Visible and InfraRed Imager | Geostationary | 2004+ |
| VIIRS | Visible Infrared Imaging Radiometer Suite | Sun-synchronous | 2012+ |
| VIRS | Visible and Infrared Scanner | Precessing | 1997-2015 |
| | Multispectral, multiangle imager/polarimeter | | |
| (A)ATSR | (Advanced) Along-Track Scanning Radiometer | Sun-synchronous | 1991-2012 |
| CHRIS | Compact High Resolution Imaging Spectrometer | Sun-synchronous | 2001+ |
| MISR | Multiangle Imaging SpectroRadiometer | Sun-synchronous | 2000+ |
| POLDER | POLarization and Directionality of the Earth's Reflectances | Sun-synchronous | 1996-1997; 2002; 2004-2013 |
| SGLI | Second-generation GLobal Imager | Sun-synchronous | 2017+ |
| SLSTR | Sea and Land Surface Temperature Radiometer | Sun-synchronous | 2016+ |
| | Nadir-looking spectrometer | | |
| AIRS | Atmospheric Infra-Red Sounder | Sun-synchronous | 2002+ |
| GOME | Global Ozone Monitoring Instrument | Sun-synchronous | 1995-2011 |
| IASI | Infrared Atmospheric Sounding Interferometer | Sun-synchronous | 2006+ |
| OMI | Ozone Monitoring Instrument | Sun-synchronous | 2004+ |
| OMPS NM | Ozone Mapping Profiler Suite Nadir Mapper | Sun-synchronous | 2012+ |
| SCIAMACHY | SCanning Imaging Absorption SpectroMeter for Atmospheric CHartographY | Sun-synchronous | 2002-2012 |
| TOMS | Total Ozone Mapping Spectrometer | Sun-synchronous | 1978-1994; 1996-2005 |
| TROPOMI | TROPOspheric Monitoring Instrument | Sun-synchronous | 2017+ |



**Table 2.** As Table 1, except for satellite instruments which have been used for aerosol extinction profiling.

| Acronym | Instrument full name | Orbit(s) | Operation period(s) |
|---|---|---|---|
| | Lidar | | |
| ALADIN | Atmospheric LAser Doppler INstrument | Sun-synchronous | 2018+ |
| CALIOP | Cloud-Aerosol LIdar with Orthogonal Polarization | Sun-synchronous | 2006+ |
| CATS | Cloud-Aerosol Transport System | Precessing | 2015-2017 |
| GLAS | Geoscience Laser Altimeter System | Polar (varied) | 2003-2010 |
| LITE | Lidar In-space Technology Experiment | Space shuttle | 1994 |
| | Limb/occultation profiler | | |
| GOMOS | Global Ozone Monitoring by Occultation of Stars | Sun-synchronous | 2002-2012 |
| MIPAS | Michelson Interferometer for Passive Atmospheric Sounding | Sun-synchronous | 2002-2012 |
| OMPS LP | Ozone Mapping Profiler Suite Limb Profiler | Sun-synchronous | 2012+ |
| OSIRIS | Optical Spectrograph and InfraRed Imaging System | Sun-synchronous | 2001+ |
| SAGE | Stratospheric Aerosol and Gas Experiment | Precessing | 1979-1982; 1984+ |
| SAM | Stratospheric Aerosol Measurement | Precessing | 1975; 1979-1993 |

details and references are provided later. Acronyms often summarise either the principle of the technique or the institution(s) which developed the algorithm. Some algorithms are not listed in this table as they do not have acronyms and are typically referred to by data producers/users by the sensor or mission name. Further, this is not an exhaustive list as numerous other approaches have been proposed in the literature; the criteria for inclusion and broader discussion in this study are that data

have been (1) processed and (2) also made generally available for scientific use. Likewise, algorithms which provide aerosol properties as a by-product but not a focus (e.g. land/ocean surface atmospheric correction approaches) are not discussed as often the aerosol components are less detailed and/or used as a sink for other error sources in the algorithm (e.g. Kahn et al., 2016).

    L2 retrieval algorithm development is typically guided by information content studies, sensitivity analyses, and retrieval

simulations to gauge which quantities a given sensor and algorithmic approach can retrieve, and with what uncertainty (e.g. Tanré et al., 1996, 1997; Hasekamp and Landgraf, 2007; Veihelmann et al., 2007; Young and Vaughan, 2009). As aerosol remote sensing is an underdetermined problem and there is considerable heterogeneity in the underlying (surface and atmospheric) conditions giving rise to the L1 signals, sensitivities and uncertainties are typically highly context-dependent. For example, retrieval of AOD from optical sensors over a dark ocean surface is typically much easier than over a bright snow-covered

surface. After an algorithm has been developed, these analyses are typically complemented by validation against reference data sets, most commonly AOD from Sun photometers such as part of the Aerosol Robotic Network (AERONET, Holben et al., 1998) over land and from hand-held instruments deployed on ocean cruises in the Maritime Aerosol Network (MAN, Smirnov et al., 2009, 2011). The resulting uncertainty estimates provided by these studies and validation analyses are *diagnostic*, i.e. for



**Table 3.** Acronyms for some aerosol retrieval algorithms/data record/institution names applied to one or more satellite instruments from Tables 1 and 2.

| Acronym | Algorithm full name |
|---|---|
| AAC | Aerosols Above Clouds |
| ADV | (A)ATSR Dual View |
| AerGOM | Aerosol profile retrieval prototype for GOMOS |
| ASV | (A)ATSR Single View |
| BAR | Bayesian Aerosol Retrieval |
| CISAR | Combined Inversion of Surface and AeRosol |
| DB | Deep Blue |
| DT | Dark Target |
| EDR | Environmental Data Record |
| ESA | European Space Agency |
| GACP | Global Aerosol Climatology Project |
| GRASP | Generalized Retrieval of Aerosol and Surface Properties |
| IMARS | Infrared Mineral Aerosol Retrieval Scheme |
| JAXA | Japan Aerospace eXploration Agency |
| LDA | Land Daily Aerosol |
| LMD | Laboratoire de Météorologie Dynamique |
| MAIAC | Multi-Angle Implementation of Atmospheric Correction |
| MAPIR | Mineral Aerosol Profiling from Infrared Radiances |
| MODACA | MODIS Above-Cloud Aerosol |
| NOAA | National Oceanic and Atmospheric Administration |
| OMACA | OMI Above-Cloud Aerosols |
| OMAERO | OMI Multi-wavelength AEROsol product |
| OMAERUV | OMI AERosol UV product |
| ORAC | Optimal Retrieval of Aerosols and Clouds |
| PMAp | Polar Multi-sensor Aerosol product |
| SOAR | Satellite Ocean Aerosol Retrieval |
| SU | Swansea University |
| SYNAER | SYNergetic AErosol Retrieval |
| ULB | Université Libre de Bruxelles |
| xBAER | eXtensible Bremen AErosol Retrieval |





a known true state they diagnose the retrieval error (difference between retrieved and true states). This is useful to identify the general tendencies for bias or loss of sensitivity under different conditions, and assess potential ways to improve on them.

Increases in the quality of instrumentation, retrieval algorithms, models, and computational power have prompted increasing desire for the provision of pixel-level uncertainty estimates in L2 aerosol data products. This has been driven in part by data
assimilation (DA) applications, which need a robust error model on data for ingestion into numerical models (Benedetti et al., 2018), often in near-real time. Diagnostic uncertainty estimates are less useful here since the true state is not known (only the retrieved state), and so a *prognostic* (predictive) uncertainty model is needed instead. Early aerosol DA applications either treated diagnostic uncertainty estimates as prognostic ones (e.g. Collins et al., 2001; Matsui et al., 2004) or constructed their own prognostic error models as part of validation and bias-correction efforts (e.g. Zhang and Reid, 2006; Benedetti et al., 2009;
Hyer et al., 2011; Shi et al., 2013). These uncertainty estimates are also valuable outside of DA to identify when a retrieval is likely to be useful for a given purpose. As an example, air quality modeling also typically uses L2 retrievals and can benefit from these uncertainties. Climate applications often use L3 aerosol data for which uncertainty estimates are yet to be robustly developed; this is an important emerging area of research regarding both methods of aggregation/reporting (e.g. Levy et al., 2009; Kinne et al., 2017; Povey and Grainger, 2019; Sayer and Knobelspiesse, 2019) and influence of sampling (e.g. Sayer
et al., 2010b; Colarco et al., 2014; Geogdzhayev et al., 2014; Schutgens et al., 2016, 2017), and L2 uncertainty estimates will be an important input to this.

Driven by these needs, many AOD data sets now provide prognostic uncertainty estimates; in some cases these additions have been developed to satisfy these user needs, while in others they have always been available as they are inherent to the retrieval technique. Unlike AOD validation, however, which has had a fairly standard methodology (Ichoku et al., 2002), there is not
yet a robust and well-used framework for evaluating these uncertainty estimates (sometimes called 'validating the validation'). This study arose from discussions as part of the international AeroSat group of aerosol remote sensing researchers, as a step toward remedying that gap. AeroSat is a grass-roots community who meet once a year, together with researchers involved in aerosol modeling (the AeroCom group) and measurement, to discuss and move toward solving common issues in the field of aerosol remote sensing. The purpose of this study is threefold:

1. To briefly review the ways in which uncertainty information has been conveyed in satellite aerosol data products (Section 2).

 2. To provide a framework for the evaluation of pixel-level AOD uncertainty estimates in satellite remote sensing, which can be adopted as a complement to AOD validation exercises going forward, and use this framework to assess AOD uncertainty estimates in several AOD retrieval products (Section 3).

3. To discuss the strengths and limitations of each these approaches, and suggest paths forward for improving the quality and use of L2 (pixel-level) uncertainty estimates in satellite aerosol remote sensing (Sections 3, 4).



## 2 Uncertainty estimates in current satellite aerosol data sets

### 2.1 Terminology

The International Standards Organization document often known as the GUM (Guide to Uncertainty in Measurement) provides standardised terminology for discussing uncertainties (Working Group 1, 2008). In the interests of standardisation and in line

with other treatments of uncertainty and error in remote sensing (e.g. Rodgers, 2000; Povey and Grainger, 2015; Loew et al., 2017; Merchant et al., 2017; Mittaz et al., 2019), the GUM terminology is also adopted here. Terms are often used inconsistently in writing or informal conversation (in particular 'error' and 'uncertainty'), so to assist the reader, definitions of relevant terms are as follows (and see previously-cited references):

– A *measurand* is a quantity to be determined (measured), in the case of this study the AOD.

– A *measurement* is the application of a technique to quantify the measurand, in this case the application of L2 retrieval algorithms to L1 satellite observations.

– The *measured value* is the output of the measurement technique, i.e. here the result of the L2 retrieval algorithm, often referred to as the 'retrieved AOD'.

– The *uncertainty* is in the general sense an expression of the dispersion of the measurand. For most of the data sets

discussed in this study it is presented as a one standard deviation ($1\sigma$) confidence interval around the retrieved value (which is defined as the *standard uncertainty* by the GUM). The true value of the measurand (AOD) is expected to lie within this confidence interval $\sim 68.4\,\%$ of the time (corresponding to one standard deviation, colloquially $1\sigma$), following Gaussian statistics.

– The *error* is the difference between the measured and true values of the measurand, i.e. here the difference between true

and retrieved AOD. Following the GUM convention, a positive error means that the measured value minus the true value is positive (and vice versa).

The error can only be known when the true value of the measurand is also known, which is rare. This is the province of validation exercises: Loew et al. (2017) note that in the remote sensing community (and adopted here), validation refers to a quality assessment of a data set, which is a different definition from that of the metrology community. While Loew et al. (2017)

omit mention of aerosols, the points discussed there are applicable to aerosol remote sensing as well. They also note that some authors (e.g. Rodgers, 2000) have adopted a stricter definition of validation to explicitly also include the question of whether theoretical characterisation and obtained properties of the data are consistent; the aforementioned 'validating the validation' framework developed in the present study is one component of this.

For validation exercises AERONET AOD data are often taken as a reference truth because the uncertainty on AERONET

AOD data (around 0.01 in the mid-visible; Eck et al., 1999) is generally much smaller than that of satellite retrievals. This enables diagnosis of retrieval errors at the times and locations of matchups with AERONET (or similar reference data), which





are often generalised to infer the likely error characteristics of retrievals under various aerosol/surface and geometric conditions. The implicit assumption is that such a generalisation is possible, but it is important to bear in mind that validation data are spatiotemporally sparse and may underrepresent or omit certain factors relative to the real world (Virtanen et al., 2018).

In contrast to error, the uncertainty can be estimated for each individual measured value (retrieval). The term 'expected error' (EE) is often used in the aerosol remote sensing literature (e.g. Remer et al., 2005; Kahn et al., 2010; Sayer et al., 2013) to define these prognostic and diagnostic estimates of the magnitude of the uncertainty, highlighting (viz. 'expected') the fact that it is a statistical quantity; in hindsight the term 'estimated uncertainty' might have been less confusing. The uncertainty is a statement about the level of confidence (expected magnitude of the error), while the actual error is a realisation drawn from the uncertainty distribution. By analogy, rolling a single unbiased die has a mean value (expectation) of 3.5 although this result is impossible to achieve on a single roll (which can take only integer values from 1-6). The various techniques which have been applied to provide prognostic estimates for AOD are discussed in Section 2.2, while Section 2.3 discusses those data sets for which only diagnostic uncertainty estimates are available. A difficulty, which this study aims to tackle, is how to tell whether these uncertainty estimates are quantitatively useful and reliable.

## 2.2 Techniques for prognostic uncertainty estimates

Examples of existing prognostic uncertainty estimates for AOD or aerosol extinction data sets are given in Table 4. These fall into two broad camps: formal error propagation techniques accounting for individual terms thought to be relevant to the overall error budget, and more empirical methods. The term 'error budget' (not defined in the GUM, but in common colloquial use) here refers to, dependent on context, the overall collection of contributions to input or output uncertainty. Strictly, one might refer instead to 'uncertainty budget' and 'uncertainty propagation', but for reader ease, the commonly-used terms are adopted here.

### 2.2.1 Formal error propagation

The formal methods which have been applied to date are in general Bayesian approaches which can be expressed in the formalism of Rodgers (2000), and are often referred to as Optimal Estimation (OE). OE approaches provide the maximum *a posteriori* (MAP) solution to the retrieval problem: maximisation of the conditional probability $P(\mathbf{x}|\mathbf{y}, \mathbf{x}_a)$ of the retrieved state vector $\mathbf{x}$, where $\mathbf{y}$, $\mathbf{x}_a$, represent the satellite measurements and any *a priori* information on $\mathbf{x}$, respectively. The MAP solution is achieved by minimisation of a cost function $J$, and the formalism allows the calculation of various contributions to the total uncertainty $\hat{\mathbf{S}}$ on the retrieved state. OE accounts for uncertainty on the satellite measurements, retrieval forward model (e.g. atmospheric/surface structure assumptions, ancillary data), *a priori* information, and smoothness constraints (on e.g. spatial, temporal, or spectral variation of parameters). While notation differs between authors (cf. Thomas et al., 2009; Dubovik et al.,

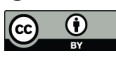

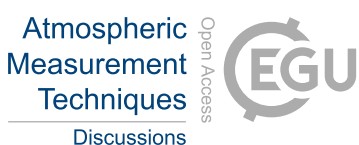

**Table 4.** AOD/extinction data sets providing prognostic uncertainty estimates, and associated key references for uncertainty estimate calculation. Where applicable, algorithm names are given first with instrument names in parentheses. See Tables 1, 2, and 3 for acronyms.

| Data set | Key references for uncertainty | Note |
| --- | --- | --- |
| ADV/ASV (ATSR2, AATSR) | Kolmonen et al. (2016); Kolmonen and Sogacheva (2018) | Jacobians at retrieval solution |
| AerGOM (GOMOS) | Vanhellemont et al. (2016) | Maximum likelihood with smoothness constraints |
| BAR (MODIS) | Lipponen et al. (2018) | Maximum likelihood, retrieves whole granule at once |
| CALIPSO | Young et al. (2013, 2018) | Propagation of contributions through lidar equation |
| CATS | Young et al. (2013) | Propagation of contributions through lidar equation |
| CISAR (CHRIS, SEVIRI) | Govaerts and Luffarelli (2018); Luffarelli and Govaerts (2019) | Optimal Estimation with smoothness constraints |
| DB AAC (MODIS, SeaWiFS, VIIRS) | Sayer et al. (2016, 2019b) | Maximum likelihood |
| DB land (MODIS) | Sayer et al. (2013) | Empirical expression from AERONET validation results |
| GOCI | Choi et al. (2018) | Empirical expression from AERONET validation results |
| GRASP (MERIS, POLDER) | Dubovik et al. (2011) | Maximum likelihood with smoothness constraints |
| IMARS (IASI) | Klüser et al. (2011) | Propagated measurement and forward model terms |
| JAXA (AHI) | Yoshida et al. (2018) | Optimal Estimation |
| LDA (SEVIRI) | Govaerts et al. (2010) | Optimal Estimation |
| LMD (AIRS, IASI) | Pierangelo et al. (2004); Capelle et al. (2014) | Parametric from sensitivity studies and validation |
| MAIAC (MODIS) | Lyapustin et al. (2018) | Propagated from uncertainty on surface reflectance |
| MAPIR (IASI) | Vandenbussche et al. (2013) | Optimal Estimation |
| MIPAS | Günther et al. (2018) | Maximum likelihood with smoothness constraints |
| MISR dark water | Witek et al. (2018b) | Width of cost function distribution vs. AOD |
| MISR heterogeneous land | Martonchik et al. (1998, 2009) | Standard deviation of well-fitting aerosol models |
| MODACA (MODIS) | Meyer et al. (2015) | Maximum likelihood |
| NOAA EDR (VIIRS) | Huang et al. (2016) | Empirical expression from AERONET validation results |
| OMPS LP | Loughman et al. (2018) | Confidence envelope based on aerosol signal strength |
| ORAC (ATSR2, AATSR, SEVIRI, SLSTR) | Thomas et al. (2009, 2010, 2017) | Optimal Estimation |
| PMAp | EUMETSAT (2015) | Standard deviation of aerosol models |
| OSIRIS | Bourassa et al. (2007); Rieger et al. (2019) | Optimal Estimation |
| SAGE | Brognier et al. (2002) | Propagated measurement plus interfering species error |
| SU (ATSR2, AATSR, MERIS+AATSR) | North et al. (2010, 2017); Bevan et al. (2012) | Second derivative of error function |
| ULB (IASI) | Clarisse et al. (2013) | Propagated measurement and forward model terms |





2011; Govaerts and Luffarelli, 2018), following Rodgers (2000) a general form of the cost function $J$ can be written

$$
\begin{aligned}
J(\mathbf{x}) = &(\mathbf{F}(\mathbf{x}) - \mathbf{y})^T \mathbf{S}_y^{-1} (\mathbf{F}(\mathbf{x}) - \mathbf{y}) \\
&+ (\mathbf{x}_a - \mathbf{x})^T \mathbf{S}_a^{-1} (\mathbf{x}_a - \mathbf{x}) \\
&+ \mathbf{x}^T \mathbf{H}_s^T \mathbf{S}_s^{-1} \mathbf{H}_s \mathbf{x} \\
&+ \dots
\end{aligned}
\tag{1}
$$

where $\mathbf{S}_y, \mathbf{S}_a$ are covariance matrices describing the measurement/forward model and *a priori* uncertainty respectively, and $\mathbf{F}(\mathbf{x})$ the forward-modelled measurements. The third term represents a generic smoothness constraint on the state vector (which

might be spatial, temporal, spectral, or otherwise) where $\mathbf{H}_s$ is a block diagonal matrix and $\mathbf{S}_s$ its associated uncertainty; the ellipsis in Equation 1 indicate the potential for expansion of $J$ to include additional smoothness terms. These covariance matrices are assumed to be Gaussian, which may not always be true in practice. Note that here $\mathbf{S}_y$ represents the total of measurement uncertainty, forward model uncertainty (due to approximations made in the radiative transfer), and the contribution of uncertainties in forward model parameters to the simulated signal at the top of atmosphere (TOA). These model parameters are

factors which affect the TOA signal but typically insignificantly enough to be retrieved. For example, many AOD retrieval algorithms ingest meteorological reanalysis to correct for the impact of absorbing trace gases (such as $H_2O$) on the satellite signal at TOA (Patadia et al., 2018), and to provide wind speed to calculate glint and whitecap contributions to sea surface reflectances (Sayer et al., 2010a). Sometimes these are represented in $J$ instead by a 'model parameter error' matrix denoted $\mathbf{S}_b$ and similar squared deviations, although mathematically since the terms in Equation 1 are additive the two formalisms are

equivalent if the model parameter uncertainty is transformed into measurement space and included in $\mathbf{S}_y$ (as is typically the case).

As $\mathbf{S}_y, \mathbf{S}_a$, etc. are square matrices, correlations between wavelengths or parameters can (and, where practical, should) be accounted for. These terms often affect several satellite bands, such that an error in e.g. reanalysis data ingested as part of an AOD retrieval would manifest in a correlated way between these bands. However, due to the difficulty in estimating these

off-diagonal elements, in practice they are frequently neglected and the covariance matrices are often assumed to be diagonal (which does not, however, mean that $\hat{\mathbf{S}}$ is diagonal). Dependent on the magnitude and sign of these correlations, their neglect can lead to over- or underestimates in the level of confidence in the solution. When the cost function has been minimised, the uncertainty $\hat{\mathbf{S}}$ on the retrieved state is given by:

$$
\hat{\mathbf{S}} = \left( \mathbf{K}^T \mathbf{S}_y^{-1} \mathbf{K} + \mathbf{S}_a^{-1} + \mathbf{H}_s^T \mathbf{S}_s^{-1} \mathbf{H}_s \right)^{-1}
\tag{2}
$$

Here $\mathbf{K}$, known as the weighting function or Jacobian matrix, is the sensitivity of the forward model to the state vector $\partial \mathbf{F}(\mathbf{x})/\partial \mathbf{x}$, typically calculated numerically. The $1\sigma$ uncertainty on the retrieved AOD is then the square root of the relevant element on the diagonal of $\hat{\mathbf{S}}$ (dependent on the contents of the state vector). Many current approaches in Table 4 omit *a priori* and/or smoothness constraints, in which case the corresponding terms in Equations 1 and 2 vanish. Only BAR and CISAR include both *a priori* and smoothness constraints. AerGOM, GRASP, and the MIPAS stratospheric aerosol data set

use smoothness constraints without *a priori* on the aerosol state. Others (LDA, JAXA AHI, MAPIR, ORAC) use *a priori* but





no smoothness constraints. Smoothness constraints are attractive for algorithms such as the GRASP application to POLDER, which includes retrieval of binned aerosol size distribution and spectral refractive index (which are expected to be smooth for physical reasons), as well as those (e.g. BAR, CISAR, GRASP) moving beyond the independent pixel approximation to take advantage of the fact that certain atmospheric/surface parameters can be expected to be spatially and/or temporally smooth on
relevant scales.

These smoothness and *a priori* constraints provide a regularisation mechanism to suppress 'noise-like' variations in the retrieved parameters when they are not well-constrained by the measurements alone, although there is a danger in that overly-strong constraints can suppress real variability. As a result, *a priori* constraints on AOD itself are often intentionally weak compared to those on other retrieved parameters. Strictly, the MAP is a maximum likelihood estimate (MLE) only if the
retrieval does not use *a priori* information, although it is often referred to as a MLE regardless (see Section 4.1 of Rodgers, 2000, for more discussion on this distinction). This distinction is made in the descriptions in Table 4.

The rest of the error propagation methods in Table 4, whether formulated as OE or not, are essentially propagating only measurement (and sometimes forward model) uncertainty through to the retrieval solution through Jacobians. MAIAC is a special case here because, rather than use the measurement uncertainty directly, it propagates the uncertainty of surface reflectance
in the 470 nm band, which is thought to be the leading contribution to the total error budget (Lyapustin et al., 2018). It is important to note that the cost function and uncertainty estimate calculations in Equation 2 are conditional on several factors:

1. The forward model must be appropriate to the problem at hand and capable of providing unbiased estimates of the observations. Typically if the forward model is fundamentally incorrect, the retrieval will frequently not converge to a solution, or have unexpectedly large $J$. For this reason, high cost values are often used in post-processing to remove
problematic pixels (e.g. undetected cloud or snow) or candidate aerosol optical models from the provided data sets (Martonchik et al., 1998; Thomas et al., 2010).

2. The covariance matrices $\mathbf{S}_y, \mathbf{S}_a, \mathbf{S}_s$ (on measurements, *a priori*, and smoothness) must be appropriate; if systematically too large or small, uncertainty estimates will likewise be too large or small. These can be tested, to an extent, by examining distributions of residuals (on measurements and *a priori*) and the cost function and comparing to theoretical
expectations (e.g. Sayer et al., 2010a, 2012c).

3. The forward model must be approximately linear with Gaussian errors near the solution. This assumption sometimes breaks down if the measurements are uninformative on a parameter and *a priori* constraints are weak or absent, and the resulting state uncertainty estimates will be invalid. This can be tested (Thomas et al., 2009; Sayer et al., 2016) by performing retrievals using simulated data, perturbing their inputs according to their assumed uncertainties, and assessing
whether the dispersion in the results is consistent with the retrieval uncertainty estimates.

4. The retrieval must have converged to the neighbourhood of the correct solution (i.e. near the global, not a local, minimum of the cost function), which can be a problem if there are degenerate solutions. In practice algorithms try to use reasonable *a priori* constraints, first guesses, and make a careful selection of which quantities to retrieve vs. which to assume (e.g.





Thomas et al., 2009; Dubovik et al., 2011). Note that the iterative method of convergence to the solution is not important in itself.

### 2.2.2 Other approaches

A particular challenge for the formal error propagation techniques is the first point above: how to quantify the individual
contributions to the error budget necessary to calculate the above covariance matrices? This difficulty has motivated some of the empirical approaches in Table 4.

Sayer et al. (2013) used the results of validation analyses against AERONET to construct an empirical relationship (discussed in more detail later) expressing the uncertainty in MODIS DB AOD retrievals as a function of various factors. This basic approach was later adopted for other data sets, including GOCI and NOAA VIIRS EDR aerosol retrievals (Huang et al., 2016;
Choi et al., 2018). This has some similarity to diagnostic EE envelopes, although importantly these prognostic estimates are framed in terms of retrieved rather than reference AOD. An advantage of this method is that, if AERONET can be taken as a truth and collocation-related uncertainty is small (Virtanen et al., 2018), it empirically accounts for the important contributions to the overall error budget without having to know their individual magnitudes. However, there are some disadvantages: if validation data are sparse or do not cover a representative range of conditions, there is a danger of overfitting the expression,
and for an ongoing data set there is no guarantee that past performance is indicative of future results as sensors age and the world changes. For a quantity without available representative validation data, the method cannot be performed. Further, programatically, it requires processing data twice: once to perform the retrievals and do the validation analysis to derive the expression, and a second time to add the resulting uncertainty estimates into the data files. The LMD IASI retrieval has a similar parametric approach (Capelle et al., 2014), although as validation data are sparse, the parametrisation draws on the
results from retrieval simulations as well.

The MISR algorithms use different approaches. Both the land and water AOD retrieval algorithms perform retrieval using each of 74 distinct aerosol optical models (known as 'mixtures') and calculate a cost function for each. In earlier algorithm versions (Martonchik et al., 1998) uncertainty was taken as the standard deviation of AOD retrieval from mixtures which fit with a cost below some threshold. This is equivalent to assuming that aerosol optical models are the dominant source of
uncertainty in the retrieval, and that the 74 mixtures provide a representative sampling of microphysical/optical properties.

This approach was refined (for retrievals over water pixels) by Witek et al. (2018b), by considering the variation of retrieval cost with AOD for each model, and transforming this to give a probability distribution of AOD, with the uncertainty taken as the width of this distribution. A similar approach has been proposed for the OMAERO retrieval by Kauppi et al. (2017), although has not yet been implemented on a large scale. It has conceptual similarities with the propagation of measurement
error in Equation 2, except calculating across the whole range of AOD state space rather than an envelope around the solution, and summing the results from multiple distinct retrievals (corresponding to the aerosol mixtures). These methods are, however, reliant on the set of available optical models being sufficient.



### 2.3 Examples of diagnostic uncertainty estimates

Available AOD data sets which do not currently provide prognostic uncertainty estimates are listed in Table 5. In these cases, algorithm papers typically summarise the results of sensitivity analyses to provide rationale for choices made in algorithm development and to provide a summary of expected performance. Sensitivity analyses often include similar aspects to those

employed in error propagation approaches: namely, characterisation of the expected effects of uncertainties in sensor calibration and forward model limitations (e.g. assumed aerosol optical models, surface reflectance) on the retrieval solution, singly or jointly. In most cases these are provided for a subset of geometries and atmosphere/surface conditions. Compared to formal error propagation, this has the advantage of being easier to communicate to a reader concerned about a particular assumption (provided the results of the sensitivity analysis are representative), but on the other hand the summary results are specific to

only the simulations performed, and real-world uncertainties may be more complicated, particularly when multiple retrieval assumptions are confounded.

Sensitivity analyses are often complemented by dedicated validation papers which summarise the results of comparisons against AERONET, MAN, or other networks (e.g. Remer et al., 2005; Kahn et al., 2010); aerosol remote sensing is fortunate compared to some other disciplines in that high-quality AOD validation data are fairly readily-available. It is common for the

results to be summarised in terms of EE envelopes or similar metrics; these envelopes are sometimes adjusted if pre-launch expectations prove too optimistic or pessimistic (e.g. Levy et al., 2013). Diagnostic and prognostic uncertainty estimates should not be regarded as exclusionary; diagnostic analysis is useful to guide algorithm refinement and assess assumptions, and many data products which provide prognostic uncertainties also show the results of diagnostic validation activities. However, extending the data sets in Table 5 to also provide prognostic estimates would improve their specificity, and utility for applications like

DA.

### 3 Statistical framework to evaluate pixel-level AOD uncertainty estimates

#### 3.1 Background and methodology

The notation adopted herein is as follows. The AOD is denoted $\tau$; unless specified otherwise, references to AOD indicate that at 550 nm. The reference (here AERONET) AOD is $\tau_A$ and satellite-retrieved AOD is $\tau_S$. The $1\sigma$ estimated uncertainties on

these are denoted $\epsilon_A$ and $\epsilon_S$ respectively. If the reference AOD is assumed to be the truth, then the error $\Delta_S$ on the satellite AOD is given by $\Delta_S = \tau_S - \tau_A$.

Figure 1 provides a simulation experiment to illustrate the relationship between AOD, uncertainty, and error distributions. The left panel is a histogram of AOD generated (1,000,000 points) assuming a Lognormal distribution with geometric mean 0.2 and geometric standard deviation 0.35, which is a typical shape for many locations in North America and Europe (O'Neill

et al., 2000). The right panel shows two distributions: in black is the distribution of the expected AOD uncertainty magnitude (often, as discussed before, called 'expected error' or EE), assuming error characteristics of the MODIS DT land retrieval, $\epsilon_S = \pm(0.05 + 0.15\tau)$ (Levy et al., 2013). This is obtained simply by multiplying the histogram in Figure 1a by the magnitude

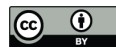



**Table 5.** AOD/extinction data sets providing sensitivity analyses and/or diagnostic uncertainty estimates, and associated key references for uncertainty. Where applicable, algorithm names are given first with instrument names in parentheses. See Tables 1, 2, and 3 for acronyms.

| Data set | Key references for uncertainty | Note |
|---|---|---|
| ALADIN | Flamant et al. (2008); Flamant et al. (2019) | Sensitivity analysis |
| DB land (AVHRR, SeaWiFS, VIIRS) | Sayer et al. (2012b, 2017, 2019a) | Envelope from sensitivity analysis/validation |
| DT land (MODIS) | Kaufman et al. (1997); Levy et al. (2013) | Envelope from sensitivity analysis/validation |
| DT water (MODIS) | Tanré et al. (1996, 1997); Levy et al. (2013) | Envelope from sensitivity analysis/validation, asymmetric |
| GACP (AVHRR) | Mishchenko et al. (1999); Geogdzhayev and Mishchenko (2015) | Sensitivity analysis, some AERONET validation |
| JAXA CAI | Fukuda et al. (2013) | Sensitivity analysis, some AERONET validation |
| JAXA GLI | Nakajima et al. (2009) | Sensitivity analysis |
| NOAA Enterprise (ABI, VIIRS) | Laszlo and Liu (2017) | Validation statistics stratified by AOD and surface type |
| NOAA ocean (AVHRR, VIRS) | Ignatov and Stowe (2000, 2002a, b); Zhao (2016) | Sensitivity analysis, some AERONET validation |
| OMACA | Torres et al. (2012); Jethva et al. (2018) | Sensitivity analysis, some airborne validation |
| OMAERO | Curier et al. (2008) | Sensitivity analysis, validation over Western Europe |
| OMAERUV | Torres et al. (1998); Ahn et al. (2013) | Envelope from sensitivity analysis/validation |
| JAXA SGLI | Mukai and Sano (2018) | Sensitivity analysis |
| SOAR (AVHRR, SeaWiFS, VIIRS) | Sayer et al. (2012a, 2017, 2018) | Envelope from sensitivity analysis/validation |
| SYNAER | Holzer-Popp et al. (2002, 2008) | Sensitivity analysis, some AERONET validation |
| TOMS | Torres et al. (1998, 2002) | Envelope from sensitivity analysis/validation |
| xBAER (MERIS) | Mei et al. (2017) | Sensitivity analysis, some AERONET validation |





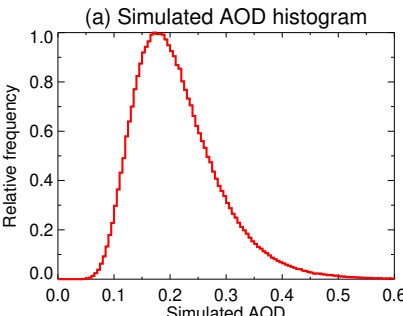
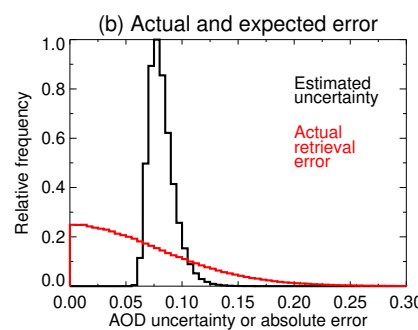

**Figure 1.** (a) Sample AOD histogram drawn from a lognormal AOD distribution with geometric mean 0.2 and geometric standard deviation 0.35. (b) Distribution of (black) estimated retrieval uncertainties and (red) actual absolute retrieval errors obtained if error characteristics followed the MODIS DT land model, $\epsilon_{\mathrm{S}} = \pm(0.05 + 0.15\tau)$.

of uncertainty $|\epsilon_{\mathrm{S}}|$. The red line, in contrast, is the distribution of actual absolute retrieval errors (i.e. $|\tau_{\mathrm{S}} - \tau_{\mathrm{A}}|$) which would be expected to be seen in a validation exercise against AERONET if the expression for $\epsilon_{\mathrm{S}}$ holds true. This red line is obtained by taking draws from the AOD distribution and then, for each, generating a Normally-distributed random number with mean 0 and standard deviation $\epsilon_{\mathrm{S}}$ to provide the retrieval error (note the absolute value of this retrieval error is shown in Figure 1b).

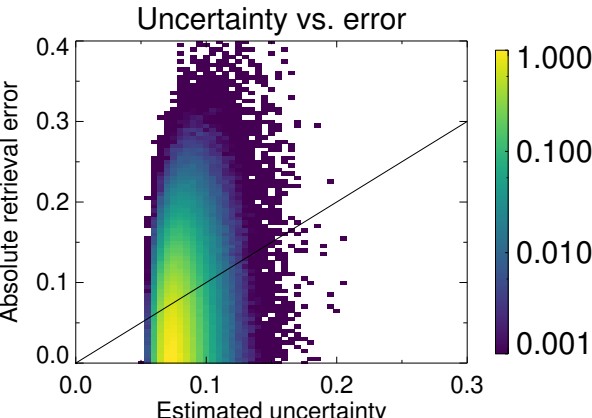

**Figure 2.** Scatter density joint histogram (on a logarithmic scale) of the simulated expected uncertainties and retrieval errors in Figure 1b. The 1:1 line is shown in black. Bins containing no data are shown in white.

5    An important nuance which bears repeating is that the distributions of estimated uncertainty and actual error in Figure 1 are quite different in shape. This is because the estimated uncertainty distribution is one of *expectations* of $\epsilon_{\mathrm{S}}$ (given the AOD distribution), while the distribution of errors is one of *realisations* of (draws from) $\epsilon_{\mathrm{S}}$. Recall again the distinction between the expectation of rolling an unbiased die (i.e. a result of 3.5) and the actual realisation (result) of rolling a die (1, 2, 3, 4, 5,





or 6). The latter distribution is broader. This illustrates why comparing errors and uncertainties on a 1:1 basis, or comparing distribution magnitudes, is not expected to yield agreement, and an evaluation of consistency requires a statistical approach. Figure 2 shows this more directly: there is little correspondence between the two on an individual basis.

When comparing satellite and reference data, the total expected discrepancy (ED) between the two for a single matchup,
denoted $\epsilon_\mathrm{T}$, should account for uncertainties on both the satellite and reference (here AERONET) data,

$$\epsilon_\mathrm{T} = \sqrt{\epsilon_\mathrm{S}^2 + \epsilon_\mathrm{A}^2}, \tag{3}$$

adding in quadrature under the assumption that the uncertainties on satellite and AERONET AOD are independent of one another. One can then define a normalised error $\Delta_\mathrm{N}$ as the ratio of the actual error to the ED, i.e.

$$\Delta_\mathrm{N} = \frac{\Delta_\mathrm{S}}{\epsilon_\mathrm{T}} = \frac{\tau_\mathrm{S} - \tau_\mathrm{A}}{\sqrt{\epsilon_\mathrm{S}^2 + \epsilon_\mathrm{A}^2}} \tag{4}$$

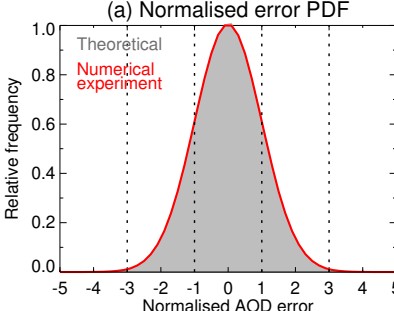
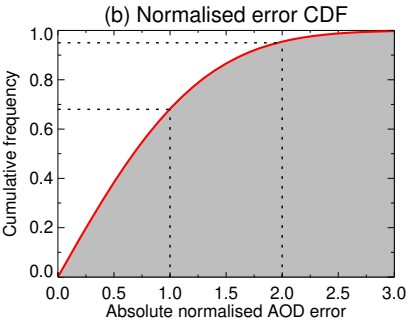

**Figure 3.** (a) PDF and (b) CDF of normalised error distributions drawn from the numerical simulations in Figure 1; theoretical (grey shading) and simulation (red) results lie on top of one another. Note the CDF is of absolute normalised error. Dashed lines indicate various well-known percentage points of Gaussian distributions.

In the ideal case $\epsilon_\mathrm{A} \ll \epsilon_\mathrm{S}$, in which case the shape of $\Delta_\mathrm{N}$ is dominated by the uncertainty and errors on the satellite-retrieved AOD. If the uncertainties on satellite and reference AOD have been calculated appropriately, and the sample is sufficiently large, then the normalised error $\Delta_\mathrm{N}$ should approximate a Gaussian distribution with mean 0 and variance 1. Thus, the distribution of $\Delta_\mathrm{N}$ can be checked in several ways against expected shapes for Gaussian distributions, for example, the probability distribution function (PDF) and cumulative distribution function (CDF) as shown in Figure 3.

The above distribution analyses are informative on the overall magnitude of retrieval errors compared to expectations (as well as, in the case of the PDF analysis, whether there is an overall bias on the retrieved AOD). However, alone they say little about the skill in assessing variations in uncertainty across the population. Taking things a step further, the data can be stratified in terms of ED and a quantile analysis performed to assess consistency with expectations. This is equivalent to taking a single location along the x-axis in Figure 2, and assessing the distribution of retrieval errors found for the points from that histogram.
These, too, should follow Gaussian statistics.





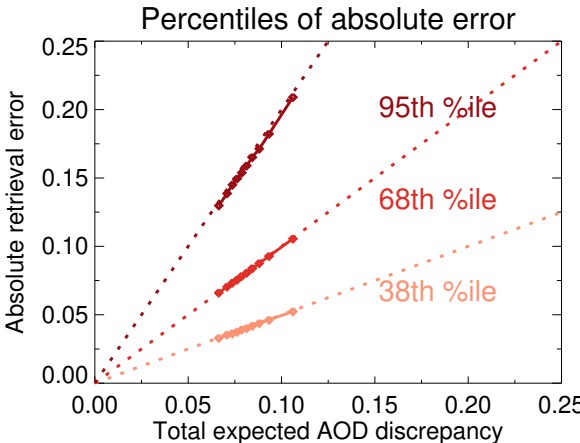

**Figure 4.** Expected AOD discrepancy against percentiles of absolute AOD retrieval error. Symbols indicate binned results from the numerical simulation; within each bin, paler to darker tones indicate the 38th, 68th, and 95th percentiles (approximate $0.5\sigma$, $1\sigma$, $2\sigma$ points) of absolute retrieval error. Dashed lines (0.5:1, 1:1, 2:1 respectively) show theoretical values for the percentiles of the same colour.

An example of this is shown in Figure 4. The data are divided by expected discrepancy $\epsilon_T$ into 10 equally-populated bins, and within each bin the 38th, 68th, and 95th percentiles (i.e. approximate $0.5\sigma$, $1\sigma$, $2\sigma$ points, following Gaussian statistics) of absolute retrieval error are plotted. If the uncertainties are appropriate, these should lie along the 0.5:1, 1:1, and 2:1 lines. This analysis provides a way of checking the validity of the uncertainty estimates across the spectrum from low to high estimated

uncertainties as opposed to population-average behaviour (i.e. do the distributions of retrieval error change in the expected way as the estimated uncertainty varies?). The 68th percentile is of most direct interest as it corresponds most directly to the expectation of the retrieval error, but examining other percentiles provides a way to assess whether the distribution is broader or narrower than expected (due to, perhaps, the presence of more or fewer outliers than expected).

The binned analysis is similar to the assessment of forecast calibration in meteorology (Dawid, 1982). Note in a forecast

sense the term *calibration* refers to a comparison of forecast vs. observed frequencies or magnitudes, distinct from the common meaning of calibration to refer to radiometric accuracy in remote sensing. By further analogy to the forecast community (cf. expressions in Murphy, 1988), a calibration skill score $s_\text{cal}$ can be defined,

$$s_\text{cal} = 1 - \frac{\sum\limits_{b=1}^{B}\left(\epsilon_{\text{T},b} - |\Delta_{\text{S},b}^{1\sigma}|\right)^2}{\sum\limits_{b=1}^{B}\left(\overline{|\Delta_\text{S}|} - |\Delta_{\text{S},b}^{1\sigma}|\right)^2},$$ (5)

where $|\Delta_{\text{S},b}^{1\sigma}|$ is the $1\sigma$ absolute retrieval error in bin $b$ (Figure 4), over $B$ bins total. This compares the observed squared

discrepancy from the 1:1 line in Figure 4 with that which would be obtained if a data user assumed that the retrieval uncertainty was equal to the mean absolute retrieval error ($\overline{|\Delta_\text{S}|}$) from a validation exercise at that location, which is what might be done in the absence of pixel-level uncertainty estimates. This skill score is computed using binned values rather than individual





matchups due to the previously-discussed nature of the relationship between uncertainty and error (Figures 1, 2). The highest possible score is 1 and a score of 0 indicates that the uncertainty estimates do not have greater skill than simply assuming the average retrieval error. If the magnitudes of $\epsilon_{\mathrm{T}}$ are in error then it is possible for $s_{\mathrm{cal}}$ to take unbounded negative values, in which case the uncertainties are said to be poorly calibrated (Dawid, 1982). This is quite a difficult test for a data set as a

positive skill score requires that both the magnitudes of the uncertainty, and the variations in both uncertainty and error, must be accurate. This may be particularly difficult if the error does not vary much at a given location. As a result $s_{\mathrm{cal}}$ should not be used as a single metric in isolation, but rather examined in a broader context.

      Figures 3 and 4 provide the basis for the framework proposed in this study. An earlier version of this method was designed during development and assessment of prognostic uncertainty estimates for MODIS DB retrievals by Sayer et al. (2013).

It has been further advanced through discussions at annual AeroSat meetings. Further practical applications of these ideas include to NOAA VIIRS AOD data by Huang et al. (2016), to GOCI data by Choi et al. (2018), and to retrievals of absorbing aerosols above clouds against airborne measurements by Sayer et al. (2019b). The idea of looking at normalised retrieval error distributions was also explored for AOD by Popp et al. (2016) and Kinne et al. (2017) when evaluating ESA Climate Change Initiative (CCI) aerosol products, and in a more general sense (with cloud top height as an example) by Merchant et al. (2017).

Indeed, the method is not restricted to AOD, although AOD has the advantage of comparatively readily-available, high-quality reference data in AERONET and other networks.

### 3.2    Practical application to satellite data products

#### 3.2.1    AERONET data used and matchup criteria

Here, the reference AOD $\tau_{\mathrm{A}}$ is provided using level 2.0 (cloud-screened and quality assured) direct-Sun data from the latest

AERONET version 3 (Giles et al., 2019). As AERONET Sun photometers do not measure at $550\,\mathrm{nm}$, the AOD is interpolated using a second-order polynomial fit to determine coefficients $a_0, a_1, a_2$ for each measurement,

$$\log(\tau_\lambda) = a_0 + a_1 \log(\lambda) + a_2 \log(\lambda)^2, \tag{6}$$

where $\lambda$ is the wavelength. All available (typically four) AOD measurements in the 440-870 nm wavelength range are used in the fit, which is more robust to calibration problems in individual channels than a bispectral approach, and accounts for

spectral curvature in $\log(\tau_\lambda)$ (Eck et al., 1999; Schuster et al., 2006). The uncertainty on mid-visible AOD is dominated by sensor calibration, and is $\sim$0.01 (Eck et al., 1999). The sampling cadence is typically once per 10 min in cloud-free, daytime conditions, but is more frequent at some sites.

      Data from a total of 12 AERONET sites, listed in Table 6, are used here to assess the AOD uncertainty estimates in various satellite data sets. This is evenly split to provide six sites to evaluate AOD retrievals from algorithms over land, and six over

water. Each category is further split; three sites are described as 'straightforward', for which the AOD retrieval problem is comparatively uncomplicated (i.e. likely no significant deviations from retrieval assumptions) and so the uncertainty estimates might be expected to be reasonable, and three sites are 'complex'. These complex sites were chosen as they have complicating



**Table 6.** AERONET sites used and their categorisation.

| Site | Latitude (° N) | Longitude (° E) | Complexity |
|---|---|---|---|
| *For land algorithm evaluation* | | | |
| Avignon | 43.93 | 4.88 | Straightforward |
| Goddard Space Flight Center (GSFC) | 38.99 | -76.84 | Straightforward |
| Palencia | 41.99 | -4.52 | Straightforward |
| Ilorin | 8.48 | 4.67 | Complex |
| Kanpur | 26.51 | 80.23 | Complex |
| Pickle Lake | 51.45 | -90.22 | Complex |
| *For water algorithm evaluation* | | | |
| Ascension Island | -7.98 | -14.41 | Straightforward |
| Midway Island | 28.21 | -177.38 | Straightforward |
| University of California Santa Barbara (UCSB) | 34.42 | -119.85 | Straightforward |
| Capo Verde | 16.73 | -22.94 | Complex |
| International Centre of Insect Physiology and Ecology (ICIPE) Mbita | -0.43 | 34.21 | Complex |
| Venise | 45.31 | 12.51 | Complex |

factors which are not well-captured by existing retrieval forward models and might be expected to lead to breakdowns in the techniques used by the retrieval algorithms to provide uncertainty estimates.

The reasons for identifying a particular site as complex are as follows. Over land, Ilorin (Nigeria) and Kanpur (India) can exhibit complicated mixtures of aerosols with distinct optical properties and vertical structure (Eck et al., 2010; Giles et al.,
2012; Fawole et al., 2016). Many AOD retrieval algorithms, in contrast, assume a single aerosol layer of homogeneous optical properties. Pickle Lake (Canada) is in an area studded by lakes of sizes similar to or smaller than satellite pixel size. This might be expected to interfere with data set land masking (which often determines algorithm choice) and surface reflectance modeling in a non-linear way (Carroll et al., 2017). Over water Capo Verde (on Sal Island, officially Republic of Cabo Verde) is characterised by frequent episodes of Saharan dust outflow; these particles have complex shapes, which are often approximated
in AOD retrieval algorithms by spheres or spheroids. This assumption leads to additional uncertainties in modeling the aerosol phase matrix and absorption cross-section, larger than for many other aerosol types, which may not be accounted for fully in the retrieval error budget (Mishchenko et al., 1997; Kalashnikova et al., 2005). ICIPE Mbita (hereafter Mbita, on the shore of Lake Victoria in Kenya) is similar to Pickle Lake but for water retrievals, i.e. it allows of sampling of nominal water pixels which may be influenced by partial misflagging of coastlines, 3D effects from the comparatively bright land, and outflow into
the water affecting surface brightness. Finally, Venise (Italy) is in the northern Adriatic Sea, slightly beyond the outflow of the Venetian lagoon, and its water colour is strongly divergent from the 'Case 1' (brightness tied to chlorophyll-a concentration; Morel, 1988) assumption employed by most AOD retrieval algorithms.





This breakdown is inherently subjective as all retrievals involve approximations; the dozen sites chosen are illustrative of different aerosol and surface regimes, but not necessarily indicative of global performance. The purpose of this study is to define and demonstrate the framework for evaluating pixel-level uncertainties, and provide some recommendations for their provision and improvement. It is hoped that, with growing acceptance of the need to evaluate pixel-level uncertainties, this

approach can be applied on a larger scale. The sites were chosen as they are fairly well-understood and have multi-year data sets (data from all available years were considered from the analysis). Note that some of the satellite data sets considered here do not provide data at some sites, for various reasons (discussed later).

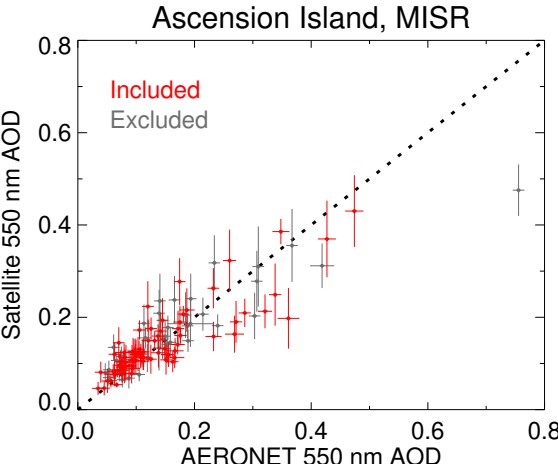

**Figure 5.** Example results of matchup and filtering criteria for MISR data at Ascension Island. Red points indicate matchups included for, and grey those excluded from, further analysis on the basis of filters described in the text. Horizontal and vertical error bars indicate the $1\sigma$ uncertainty on AERONET and MISR data, respectively. The 1:1 line is dashed black.

The matchup protocol is as follows. AERONET data are averaged within $\pm 15\,\mathrm{min}$ of each satellite overpass (providing $\tau_\mathrm{A}$), and compared with the closest successful satellite retrieval which has a pixel centre within $10\,\mathrm{km}$ of the AERONET site. This

provides $\tau_\mathrm{S}$ and $\epsilon_\mathrm{S}$. Each satellite data set's recommended quality assurance (QA) filtering criteria are applied as provided in the data products. The AERONET uncertainty, $\epsilon_\mathrm{A}$, is taken as the quadrature sum of the AERONET measurement uncertainty ($\pm 0.01$; Eck et al., 1999) and standard deviation of the AERONET measurements (typically 2-3) during the $\pm 15\,\mathrm{min}$ temporal window. Additionally, matchups are discarded if $\epsilon_\mathrm{A} > 0.02$ or if only one AERONET measurement is obtained during the time window, as this indicates the potential for heterogeneous scenes. Dependent on site and sensor, this additional filtering step

removes around 10-60 % of potential matchups; Figure 5 shows an example for MISR over-water retrievals at the Ascension Island site. As a reminder, the focus here is not on validating the AOD, but rather validating the AOD uncertainty estimates (vertical lines in the Figure).

These matchup criteria are stricter than is commonly applied for AOD validation (e.g. Ichoku et al., 2002), which typically averages AERONET data within $\pm 30\text{-}60\,\mathrm{min}$ and satellite retrievals within $\sim \pm 25\,\mathrm{km}$; the smaller spatiotemporal window





and additional filtering critera decrease the potential (unknown) contribution of collocation uncertainty to $\epsilon_A$, which increases as the colocation criteria are loosened (Virtanen et al., 2018). The reasoning behind taking the nearest, rather than average, satellite retrieval is similar: averaging would have the potential to decrease the apparent retrieval error, which would make the comparison less useful for evaluating $\epsilon_S$. Weakening these criteria could increase the data volume for analysis at the expense

of increased colocation-related uncertainty and there is no objective way to determine universal optimal thresholds.

This work considers satellite AOD products from seven algorithm teams; five of these contain both land and water retrievals (albeit sometimes with different algorithms), while two only cover land retrievals. Only pixels retrieved as land are used for comparison with AERONET data from land sites in Table 6, and vice-versa for water sites. These data sets are briefly described below and the reader is referred to references cited here and in Tables 4 and 5 for additional information. Note in the discussion

the term 'pixel' refers to individual L2 retrievals, sometimes referred to 'superpixels' in the literature as they are often coarser than the source L1 data.

### 3.2.2 MODIS data sets

Four of the data sets (three land, one water) are derived from MODIS measurements; there are two MODIS sensors, providing data since 2000 and 2002 on the Terra and Aqua satellites, respectively. The sensors have a 2,330 km swath width which is

advantageous in providing a large data volume for analysis. Since launch, the MODIS aerosol data products have included AOD from the DT algorithm family, which has separate algorithms for water and vegetated land pixels (Levy et al., 2013). These data sets provide only diagnostic uncertainty estimates of the form $\epsilon_S = \pm(a + b\tau_A)$; in practice (and here) these are often treated as if they were framed instead in terms of $\tau_S$ with the same coefficients $a, b$ when a prognostic estimate is needed. For retrievals over land, $\epsilon_S = \pm(0.05 + 0.15\tau_A)$, which is consistent with the expected performance of the algorithms at launch

(Remer et al., 2005). Over water, the estimate has been revised since launch to $\epsilon_S = \pm(0.03 + 0.1\tau_A)$. Limited validation based on Collection 6 data by Levy et al. (2013) suggested that there might be an asymmetry to the envelope with the $1\sigma$ range over water being from $-0.02 - 0.1\tau_A$ to $+0.04 + 0.1\tau_A$. This has not yet been corroborated by global validation of C6 or the latest Collection 6.1 (C6.1), and it is also plausible that calibration updates in C6.1 may have ameliorated some of this bias. As a result the symmetric envelope is used here.

The DB algorithm retrieves AOD only over land and was introduced to fill gaps in DT coverage due to bright surfaces such as deserts (although has since been expanded to include vegetated land surfaces as well). The latest version is described by Hsu et al. (2019). Prognostic AOD retrieval uncertainties are estimated as described in Sayer et al. (2013),

$$\epsilon_S = \pm\left(\frac{a + b\tau_S}{\frac{1}{\mu_0} + \frac{1}{\mu}}\right), \tag{7}$$

where $\mu_0, \mu$ are the cosines of solar and view zenith angles, respectively, and $a, b$ coefficients depending on QA flag value,

sensor, and (since C6.1) surface type. The latest values of $a, b$ are given by Hsu et al. (2019).

BAR also performs retrievals only over land; it uses the same radiative transfer forward model as DT, but reformulates the problem to retrieve the MAP solution of aerosol properties and surface reflectance simultaneously for all vegetated pixels in a single granule (Lipponen et al., 2018). This includes both *a priori* information and spatial smoothing constraints. Uncertainty





estimates are provided organically by the MAP technique (Equation 2). Note BAR data are only available at present from 2006-2017.

For all MODIS products, data from the latest C6.1 are used. All products are provided at nominal (at-nadir) $10\,\mathrm{km}$ horizontal pixel size. Identical algorithms (and approaches for estimating uncertainty) are applied to both Terra and Aqua measurements, and the results of the evaluation were not distinguishable for Terra and Aqua data. For conciseness and to increase data volume Terra and Aqua data are not separated in the discussion going forward.

### 3.2.3 MISR data sets

The MISR sensor also flies on the Terra platform, and consists of 9 cameras viewing the Earth at different angles, with a fully-overlapped swath width around $380\,\mathrm{km}$ (Diner et al., 1998). The latest version 23, used here, provides AOD retrievals at $4.4\,\mathrm{km}$ horizontal pixel size. Both land and water retrievals (Garay et al., 2017; Witek et al., 2018b) attempt retrieval using each of 74 candidate aerosol mixtures, although they differ in their surface reflectance models and uncertainty estimates. The over-land 'heterogeneous surface' retrieval estimates uncertainty as the standard deviation of AOD retrieved using those aerosol mixtures which provide a sufficiently close match to TOA measurements (Martonchik et al., 1998, 2009). The 'dark water' approach (Witek et al., 2018b) looks at the variation of a cost function across the range of potential AOD and aerosol mixtures,

$$f(\tau) = \frac{1}{N} \sum_{m=1}^{N} \frac{1}{\chi_m^2(\tau)}, \tag{8}$$

where the sum is over $N = 74$ aerosol mixtures and $\chi_m^2$ is a cost function similar to the first term of Equation 1. The uncertainty $\epsilon_S$ is then taken as the full-width at half maximum of $f(\tau)$, which is often found to be monomodal and close to Gaussian (Witek et al., 2018b). Note that MISR does not provide retrievals over Mbita or Venise as the 'dark water' algorithm logic excludes pixels within the matchup radius used here as too bright and unsuitable; thus, the approach cannot be evaluated at those sites.

### 3.2.4 ATSR data sets

The ATSRs were dual-view instruments, measuring near-simultaneously at nadir and near $55°$ forward. ATSR2 (1995-2003) and AATSR (2002-2012) had four solar and three infrared bands, with approximately $1\,\mathrm{km}$ pixel sizes, and a $550\,\mathrm{km}$ swath (although ATSR2 operated in a narrow-swath mode over oceans). Their predecessor ATSR1 lacked three of the solar bands and so has not been used widely for AOD retrieval. In 2016 the first of a new generation of successor instruments (the SLSTRs) was launched; SLSTR has several additional bands, a rear view instead of forward, the native spatial resolution of solar bands is finer, and the swath broader (Coppo et al., 2010). This study uses two data sets derived from this family of sensors.

ORAC is a generalised OE retrieval scheme which has been applied to multiple satellite instruments. Here, the version 4.01 ATSR2 and AATSR from the ESA CCI are used (Thomas et al., 2017), along with an initial version 1.00 of data from SLSTR. ORAC provides AOD retrievals over both land and ocean surfaces; the retrieval approaches are the same except for the surface reflectance models, which also inform the *a priori* and covariance matrices. Over water, surface reflectance is modelled according to Sayer et al. (2010a) with fairly strong *a priori* constraints. Over land, two approaches have been implemented





in ORAC; the one used here is a model developed initially for the SU (A)ATSR retrieval algorithm (North et al., 1999) which assumes that the ratio between forward and nadir surface reflectance is spectrally invariant, and has very weak *a priori* constraints. Note that AOD and aerosol effective radius have weak and strong *a priori* constraints, respectively. Retrievals are performed at native resolution, and cost functions and uncertainty estimates are as in Equations 1 and 2 without smoothness

constraints. ORAC simultaneously retrieves aerosol and surface properties, peforming an AOD retrieval for each of a number (here, 10) candidate aerosol optical models (mixing four components defined by the aerosol CCI; Holzer-Popp et al., 2013), and choosing the one with the lowest cost as the most likely solution. Retrievals passing quality checks (Thomas et al., 2017) are then averaged to a $10\,\mathrm{km}$ Earth-referenced sinusoidal grid.

    ADV uses the ATSR dual-view over land to retrieve the contribution to total AOD from each of three aerosol CCI components

(with the fraction of the fourth dust component prescribed from a climatology) by assuming that the ratio of surface reflectance between the sensor's two views is spectrally flat. This has some similarity with the North et al. (1999) approach, except for ADV the ratio is estimated from observations in the $1600\,\mathrm{nm}$ band where the atmosphere is typically most transparent, rather than being a freely-retrieved parameter (Kolmonen et al., 2016). Over the water, the algorithm only uses the instruments' forward view as this has a longer atmospheric path length and is less strongly affected by Sun glint. Because of this, the

water implementation is often called ASV rather than ADV (Table 3), although for convenience here the term ADV is used throughout. Water surface reflectance is modelled as a combination of Fresnel reflectance and the cholorphyll-driven model of Morel (1988). The land and water algorithms treat other factors (e.g. aerosol optical models) in the same way. Unlike ORAC, ADV aggregates to a $10\,\mathrm{km}$ grid before performing the retrievals. ADV uncertainty estimates are calculated using Jacobians at the retrieval solution, i.e. the first component of Equation 2, with $\mathbf{S}_y$ assumed diagonal. The uncertainty on the

TOA measurements is taken as $5\,\%$, which is somewhat larger than that assumed by ORAC, so ADV is implicitly adding some forward model uncertainty into this calculation. Version 3.11 of the data sets (Kolmonen and Sogacheva, 2018), also from the ESA aerosol CCI, are used here.

    Aside from pixel/swath differences, for both ADV and ORAC the implementation of the algorithms is the same for the three sensors. Matchups from the two (for ADV) or all three (for ORAC) sensors are combined here in the analysis, to increase

data volume, due to the similarity in sensor characteristics and algorithm implementation. Note however that the difference in viewing directions between (A)ATSR and SLSTR (i.e. forward vs. rear) means different scattering angle ranges are probed over the two hemispheres, which influences the geographic distributions of retrieval uncertainties. For both of these data sets, a large majority of matchups ($75\,\%$ or more) obtained are with AATSR, as the ATSR2 mission ended before the AERONET network became as extensive as it is at present, and the SLSTR record to date is short. The results do not significantly change

if only AATSR data are considered.

### 3.2.5   CISAR SEVIRI

Unlike the other data sets considered here, the SEVIRI sensors fly on geostationary rather than polar-orbiting platforms. This analysis uses data from the first version of the CISAR algorithm (Govaerts and Luffarelli, 2018) applied to SEVIRI aboard Meteosat 9; due to computational constraints, only SEVIRI data from 2008-2009 have been processed and included here. This





sensor has a sampling cadence of 15 min and observes a disk centred over North Africa, covering primarily Africa, Europe, and surrounding oceans. The horizontal sampling distance is 3 km at nadir, increasing to around 10 km near the limits of useful coverage. This sampling means that several of the AERONET sites (GSFC, Kanpur, Midway Island, Pickle Lake, UCSB) are not seen by the sensor and cannot be analysed.

5   CISAR is also an OE retrieval scheme, which in its SEVIRI application accumulates cloud-free measurements from three solar bands over a period of five days and simultaneously retrieves aerosol and surface properties, reporting at each SEVIRI timestep. Surface reflectance is modelled following Rahman et al. (1993) over land and Cox and Munk (1954a, b) over water, although the retrieval approach is otherwise the same between the two surface types. It employs *a priori* data and several smoothness constraints, and so uncertainty estimates (Luffarelli and Govaerts, 2019) broadly follow Equation 2.

10 **3.3   Results**

**Table 7.** Number of matchups obtained for each AERONET site and data set, together with climatological cloud fraction.

| Data set | AERONET site | | | | | |
|---|---|---|---|---|---|---|
| | Land matchup counts | | | | | |
| | Avignon | GSFC | Palencia | Ilorin | Kanpur | Pickle Lake |
| ADV | 266 | 199 | 98 | 89 | 100 | 57 |
| BAR | 1793 | 2088 | 2017 | 893 | 1087 | 1119 |
| CISAR | 1749 | 0 | 868 | 493 | 0 | 0 |
| DB | 3045 | 3010 | 1924 | 1144 | 1493 | 1068 |
| DT | 2519 | 2409 | 1774 | 895 | 1250 | 529 |
| MISR | 241 | 271 | 203 | 82 | 153 | 108 |
| ORAC | 344 | 326 | 200 | 105 | 104 | 106 |
| Cloud fraction | 0.50 | 0.57 | 0.55 | 0.68 | 0.55 | 0.67 |
| | Water matchup counts | | | | | |
| | Ascension Island | Midway Island | UCSB | Capo Verde | Mbita | Venise |
| ADV | 30 | 43 | 81 | 59 | 57 | 137 |
| CISAR | 210 | 0 | 0 | 716 | 336 | 1442 |
| DT | 748 | 443 | 1812 | 768 | 341 | 2698 |
| MISR | 74 | 59 | 196 | 115 | 0 | 0 |
| ORAC | 66 | 79 | 135 | 143 | 68 | 257 |
| Cloud fraction | 0.59 | 0.63 | 0.34 | 0.72 | 0.42 | 0.58 |

With the above criteria, the number of matchups $n$ obtained for each AERONET site with each data set is shown in Table 7. This additionally includes the long-term climatological mean (March 2000-February 2019) daytime cloud fraction $f_C$ from MODIS Terra, taken from the C6.1 level 3 monthly product (MOD08_M3) for the $1°$ grid cell in which the AERONET site





lies. The cloud masking approach is described by Frey et al. (2008), with more recent updates listed in Section 3 of Baum et al. (2012). Data from Terra are used as the majority of the aerosol data sets, like Terra, have a late-morning overpass time.

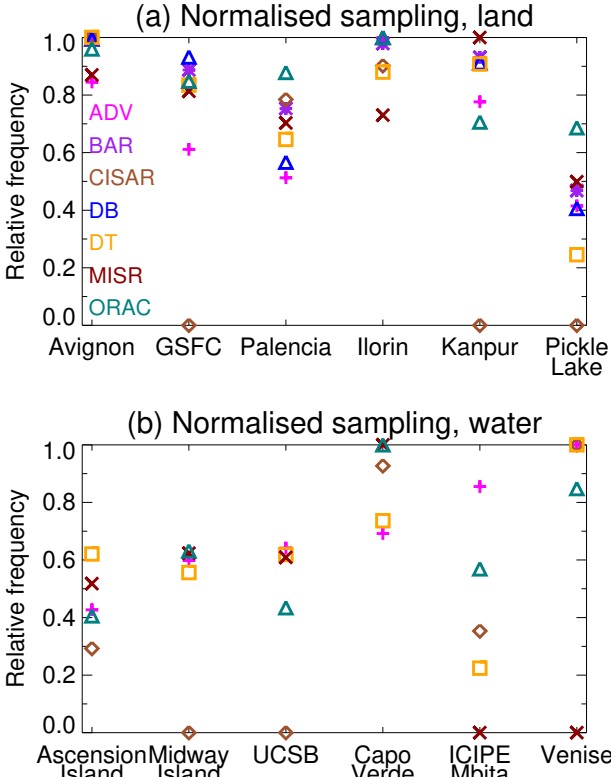

**Figure 6.** Site-to-site corrected sampling $\hat{n}$ for each data set, shown on a relative scale. Symbols are used to aid in differentiating overlapping data points but carry no further information.

To make the counts more comparable between sites a sampling-corrected count $\hat{n}$ can be calculated,

$$\hat{n} = n \frac{\cos(\phi)}{(1 - f_C)} \frac{m_S}{m_A}, \qquad (9)$$

5 where $\phi$ is the site's latitude (important as for polar-orbiting satellites a given latitude is overflown proportional to $1/\cos(\phi)$), $m_S$ the number of months of the satellite record, and $m_A$ the number of months during the satellite record for which the AERONET site was in operation. For example, CISAR data used here cover the period 2008-2009 ($m_S = 24$); for these years, AERONET data at Ascension Island are available for 5 months of 2008 and 11 of 2009 ($m_A = 16$). Equation 9 thus provides a first-order estimate of the number of matchups which would have been obtained in the absence of clouds (as the data sets

10 consider cloud-free pixels only), an equal rate of being overflown, and with the AERONET site in constant operation through the satellite lifetime. Normalising each satellite data set to the maximum of $\hat{n}$ across sites (to account for swath width and mission length differences, which determine total counts) provides a relative measure of how often each data set provides a





valid retrieval at each location; the resulting relative sampling frequencies are shown in Figure 6. This measure will be used in the ongoing discussion. Note that as CISAR is applied to geostationary SEVIRI data, the factor of $\cos(\phi)$ is omitted (since each point on the disk is sampled once per scan, and each point outside the disk is never seen).

**Table 8.** Calibration skill scores $s_{\mathrm{cal}}$ and coefficient of determination $R^2$ from binned $1\sigma$ uncertainties in Figures 7 and 8.

| Data set | AERONET site | | | | | | | | | | | |
|---|---|---|---|---|---|---|---|---|---|---|---|---|
| | $s_{\mathrm{cal}}$ | $R^2$ | $s_{\mathrm{cal}}$ | $R^2$ | $s_{\mathrm{cal}}$ | $R^2$ | $s_{\mathrm{cal}}$ | $R^2$ | $s_{\mathrm{cal}}$ | $R^2$ | $s_{\mathrm{cal}}$ | $R^2$ |
| Land calibration skill scores/$R^2$ | | | | | | | | | | | | |
| | Avignon | | GSFC | | Palencia | | Ilorin | | Kanpur | | Pickle Lake | |
| ADV | <0 | 0.58 | <0 | 0.87 | <0 | 0.72 | 0.29 | 0.99 | <0 | 0.99 | 0.08 | 0.99 |
| BAR | <0 | 0.55 | <0 | 0.94 | <0 | 0.57 | <0 | 0.37 | <0 | 0.84 | 0.11 | 0.88 |
| CISAR | <0 | 0.67 | - | - | <0 | 0.075 | <0 | 0.94 | - | - | - | - |
| DB | <0 | 0.98 | 0.57 | 0.98 | 0.65 | 0.96 | <0 | 0.99 | <0 | 0.97 | 0.61 | 0.86 |
| DT | 0.57 | 0.89 | 0.47 | 0.89 | 0.53 | 0.92 | <0 | 0.04 | 0.69 | 0.91 | 0.50 | 0.99 |
| MISR | 0.84 | 0.85 | 0.97 | 0.99 | 0.93 | 0.98 | 0.62 | 0.75 | <0 | 0.38 | 0.87 | 0.96 |
| ORAC | <0 | 0.82 | <0 | 0.70 | <0 | 0.36 | 0.82 | 0.95 | <0 | 0.88 | <0 | 0.84 |
| Water calibration skill scores/$R^2$ | | | | | | | | | | | | |
| | Ascension Island | | Midway Island | | UCSB | | Capo Verde | | Mbita | | Venise | |
| ADV | <0 | - | <0 | - | 0.40 | 0.79 | <0 | 0.91 | <0 | 0.35 | 0.11 | 0.70 |
| CISAR | <0 | 0.11 | - | - | - | - | <0 | 0.42 | <0 | 0.28 | <0 | 0.33 |
| DT | 0.72 | 0.87 | 0.38 | 0.95 | 0.45 | 0.99 | 0.73 | 0.93 | 0.62 | 0.92 | 0.63 | 0.98 |
| MISR | 0.52 | 0.94 | <0 | 0.45 | 0.80 | 0.97 | 0.78 | 0.94 | - | - | - | - |
| ORAC | <0 | 0.84 | <0 | 0.48 | <0 | 0.06 | <0 | 0.92 | <0 | 0.047 | <0 | 0.063 |

Graphical evaluation of the pixel-level uncertainties are shown in Figures 7 and 8, for land and water retrievals respectively. In both of these the left-hand column shows CDFs of absolute normalised error $|\Delta_{\mathrm{N}}|$ against theoretical expectations (cf. Figure 3b), and the middle and right columns show the ED $\epsilon_{\mathrm{T}}$ and twice ED, binned, against the $1\sigma$ and $2\sigma$ points of absolute retrieval error $|\Delta_{\mathrm{S}}|$, respectively (cf. Figure 4). Due to the very different sampling between data sets and sites (Table 7), the number of bins is taken as the lesser of $n/20$ or $n^{1/3}$ (rounded to the nearest integer). This choice is a balance between well-populated bins to obtain robust statistics, and the desire to examine behaviour across a broad range of $\epsilon_{\mathrm{T}}$. These Figures also include an estimate of the digitisation uncertainty on the binned values: for example, in a bin containing 100 matchups, the uncertainty on the 68th percentile ($1\sigma$ point) binned value shown is taken as the range from the 67th to 69th matchup in the bin. For the MODIS-based records (which have the highest sampling) this digitisation uncertainty is often negligible, but for others (ADV, MISR, ORAC) it is sometimes not.

Together, these enable a simple evaluation of the pixel-level uncertainty estimates: the CDFs in Figures 7 and 8 assess the overall magnitude of normalised errors and shape of the distribution, while the binned ED assesses the overall skill in



**Figure 7.** Evaluation of pixel-level uncertainty estimates for over-land retrievals. Each row corresponds to a different AERONET site, and colours are used to distinguish data sets. The left-hand column shows a CDF of the absolute normalised retrieval error $|\Delta_N|$ (cf. Figure 3b) and the middle and right columns show $1\sigma$ and $2\sigma$ expected discrepancy ED vs. absolute retrieval errors $|\Delta_S|$ (cf. Figure 4) respectively. In the left column, theoretical expectations are shaded grey; in the others, the 1:1 line is indicated dashed in grey, and vertical bars indicate the uncertainty on the bin value, as described in the text.

**Figure 8.** As Figure 7, except for AERONET sites used for over-water retrieval evaluation.





specificity of the estimates. In all of these Figures, the sites are grouped in triplets according to whether they were expected to be straightforward or complicated test cases for the uncertainty estimate techniques (Table 6). Table 8 provides the overall calibration skill scores for $1\sigma$ error at each site (Equation 5), plus the coefficient of determination $R^2$ (where at least 3 bins were available) between binned uncertainty and $1\sigma$ error from the middle columns of Figures 7 and 8.

### 3.3.1 Land sites

Turning to the land sites (Figure 7), all the techniques show some skill in that the ED generally increases with retrieval error. There is, however, considerable variation between sites (which points to the utility of considering results site-by-site for this demonstration analysis) and data sets. For the 'straightforward' sites, there is an overall tendency for the uncertainty estimates to be too large. This may indicate that the retrieval error budgets are a little too pessimistic; since overall errors and uncertainties

also tend to be small at these sites, it is also possible that the uncertainty on the AERONET data (which can be a non-negligible contribution to ED here) is overestimated. A notable exception here is MISR, for which uncertainty estimates are very close to theoretical expectations. This implies that the overall assumptions made by this technique (that the principle contribution to error is in aerosol optical model assumptions, and the 74 mixtures provide a representative set, such that the standard deviation of retrieved AOD between well-fitting mixtures is a good proxy for uncertainty) is valid. A second exception is CISAR, which

more significantly overestimates the uncertainty, indicating that the retrieval is more robust than expected. For these sites the binned plots of $1\sigma$ and $2\sigma$ retrieval error vs. ED look similar, suggesting that, within each bin, the retrieval errors are roughly Gaussian (even if the magnitudes of uncertainty are not perfectly estimated). MODIS DT tends to overestimate uncertainty on the low end and underestimate on the high end, suggesting (at least for these sites) that the first and second coefficients in the expression $\epsilon_S = \pm(0.05 + 0.15\tau)$ may need to be decreased and increased, respectively.

For the 'complex' land sites, the picture is different. At Ilorin, MODIS DB and ADV tend to overestimate uncertainty while the others underestimate. This site was chosen as a test case because of the complexity of its aerosol optical properties, which are more absorbing than assumed by many retrieval algorithms and can show large spatiotemporal heterogeneity due to a complex mix of sources (Eck et al., 2010; Giles et al., 2012; Fawole et al., 2016). Using aircraft measurements, Johnson et al. (2008) found mid-visible single scattering albedo (SSA) from smoke-dominated cases between 0.73-0.93, with a central

estimate for the smoke component of 0.81. DB has a regional SSA map with more granularity (Hsu et al., 2019), while the other algorithms do not contain sufficiently absorbing particles, leading to a breakdown in their uncertainty estimates when strong absorption is present.

The most absorbing component in the MISR aerosol mixtures has an SSA of 0.80 at 558 nm; mixtures including this component have SSA from 0.81-0.96, and all other MISR mixtures have SSA>0.90 (Tables 2, 3 of Kahn et al., 2010). In

smoke cases retrievals are biased low and the uncertainty estimates too narrow because the set of candidate aerosol mixtures is not representative of optical properties at this location. MODIS DT and BAR (which uses the same optical models as DT) assume a fine-mode dominated model with mid-visible SSA of 0.85 from December-May and 0.90 from June-November (Figure 3 of Levy et al., 2007), and mix this with a less absorbing coarse-dominated model, so suffer similar issues. CISAR retrieves AOD by a combination of aerosol vertices in SSA-asymmetry parameter space; the most absorbing (for SEVIRI's





640 nm band, which is the shortest wavelength) has SSA around 0.79 (Figure 4 of Luffarelli and Govaerts, 2019); due to the spectral curvature of smoke SSA, this would imply a weaker effective absorption in the mid-visible. ADV and ORAC share aerosol components prescribed by the aerosol CCI (Holzer-Popp et al., 2013); the most absorbing fine-mode component has mid-visible SSA around 0.80, although this is also always mixed with more weakly-absorbing fine-mode (which have SSA of

0.98) and coarse-mode particles in varying proportions, so in practice the assumed SSA is always higher (Tables 1 and 2 of Thomas et al., 2017). It may be that ADV is providing reasonable estimates at this site despite this, due to is somewhat larger assumed forward model uncertainty than ORAC. For Kanpur, except for MISR (which has similar issues to Ilorin) and CISAR (as SEVIRI does not observe the site) these issues are lessened. This may be because, while Kanpur has similar complex mixed aerosol conditions, the components are overall less strongly absorbing and so these issues are less acute, with a typical SSA

(similar to that of Ilorin in mixed, as opposed to smoke-dominated, conditions) around 0.89 (Giles et al., 2012). The issues with MISR may imply the wrong mixture(s) are being selected here.

The case at Pickle Lake is more diverse: similar to the 'straightforward' sites MODIS DT, DB, and BAR all overestimate uncertainty. ADV and MISR are fairly close to theoretical values; despite this, their skill scores are fairly low (Table 8) as the magnitudes of their uncertainties are not perfect and the range of $1\sigma$ retrieval errors is fairly small. All these algorithms

provide retrievals significantly less often than would be expected by the site's cloud cover, latitude, and AERONET availability (Figure 6). This implies that the algorithms may be coping with a potential violation of assumptions (i.e. land mask issues from numerous small lakes) by simply not providing a retrieval at all. ORAC underestimates uncertainties at this site, but provides retrievals relatively more frequently than the other data sets. As the land/sea mask is determined at full (1 km) resolution and used to set the surface model, it is likely that some of the pixels within the 10 km grid will be affected by mis-flagging/mixed

surface issues, contributing to additional errors which are not being caught by these quality checks. Which behaviour is more desirable (no data vs. more uncertain data than expected) is a philosophical and application-dependent matter. As it lies outside the SEVIRI disk, CISAR provides no retrievals at this site.

Aside from DB, DT, and MISR, skill scores (Table 8) are in most cases negative; for the former two the uncertainty estimates are somewhat empirical and not independent of the AERONET data, so the fact they are fairly well-calibrated is not surpris-

ing. Despite this $R^2$ is typically not negligible (although the small number of bins means the estimates of $R^2$ are somewhat uncertain). This implies that, while the absolute magnitudes of estimated uncertainty are often too small/large, the techniques do show some skill at predicting which retrievals are comparatively less/more uncertain at a variety of locations. Neither $s_{\mathrm{cal}}$ nor $R^2$ should be overinterpreted in terms of site-to-site variations, as these depend strongly on the number of bins, range in estimated uncertainties, and range in actual retrieval errors at a given site. The main points of note are whether $s_{\mathrm{cal}} > 0$, and

whether there is a positive association between binned uncertainty and error.

### 3.3.2  Water sites

For the water sites (Figure 8), only five satellite data sets are available–recall also that the MODIS DT uncertainty envelope is narrower than over land, and the MISR uncertainty is a PDF based on a cost function composited over AOD and aerosol mixtures rather than (as over land) a simple standard deviation. At the 'straightforward' sites there is some commonality with





the land sites. Specifically, the MISR approach works fairly well, CISAR overestimates uncertainty (although of the three, only Ascension Island is within the SEVIRI disk), and MODIS DT slightly overestimates uncertainty overall, with a tendency to overestimate on the low end and underestimate on the high end. In general a similar picture is also seen in terms of $s_{cal}$ and $R^2$: most data sets are not well-calibrated, although there is skill at assessing variations in uncertainty at individual sites.

ADV and ORAC are more systematic in their underestimation of uncertainty over water compared to over land, although as the over-water errors are often fairly small in absolute terms, they appear fairly large in relative terms. This difference in the ATSR-based records between land and ocean sites is intriguing. ADV assumes 5 % uncertainty in the TOA signal while ORAC includes separate measurement and forward model terms for a slightly lower total uncertainty overall (typically 3-4 % dependent on band and view), which in part explains ORAC's larger normalised errors. The common behaviour either implies
(1) that the calibration of the sensors may be biased or more uncertain than expected for these fairly dark ocean scenes, or (2) that the over-water surface reflectance models or (for ORAC) their uncertainties (either in their contribution to forward model error in $\mathbf{S}_y$, or the strength of the *a priori* constraint in $\mathbf{S}_a$) might be less reliable than assumed. A thorough comparison between the two data sets using the matchups collected here is difficult due to the fairly low data volumes involved, especially for ADV. ADV provides significantly fewer retrievals overall than ORAC (for both land and water), implying stricter pixel
selection/retention criteria; this is consistent with ESA CCI validation analysis of earlier versions of these data sets by Popp et al. (2016) and Kinne et al. (2017).

    Despite the expected complexities at Capo Verde from mixtures of low-level sea spray and higher-altitude nonspherical mineral dust (Mishchenko et al., 1997; Kalashnikova et al., 2005), the error characterisation at this complex site does not appear different from that obtained at the more straightforward sites. Interestingly, these algorithms seem more selective about
when to provide retrievals at the three straightforward sites than they are at Capo Verde (Figure 6). The reasons for this are unclear unless the estimate provided by $\hat{n}$ (Equation 9) is not a good approximation for these sites; each is close to the coast and all should be roughly equally affected by Sun-glint sampling-related losses.

    Mbita is in some sense the inverse of the land site Pickle Lake, and similar comments apply. MODIS DT uncertainties are reasonable, although the data volume is fairly low relative to expectations from Figure 6. ADV and ORAC retrieve more
frequently, and perform well but with more high-error outliers than expected, likely due to mixed or misflagged land/water pixels. CISAR retrieves with a similar frequency at Mbita to Ascension Island (that is, less than expected, but no less so than at the straightforward site). Looking at the binned ED vs. error, the errors for the $1\sigma$ points (Figure 8n) are slightly overestimated and that for the $2\sigma$ points (Figure 8o) underestimated, implying more extreme outliers than expected, indicating possible surface contamination issues. Note MISR does not provide retrievals at this site as the algorithm does not consider Lake Victoria to be
dark water.

    Venise is sampled close to the expected rates by ADV, CISAR, MODIS DT, and ORAC (Figure 6), and again excluded by MISR due to the bright, turbid water. Here, the CISAR $1\sigma$ retrieval error is ∼0.05 and the $2\sigma$ error is about double that, regardless of the ED, and the uncertainty estimates do not show skill overall. As SEVIRI's wavelengths (640, 810, 1640 nm) are less strongly affected by water turbidity than the other sensors, the issues causing complexity here may not apply and the
overall tendency for CISAR to report too large an uncertainty may be dominating. ADV and DT results are reasonably in line





with expectations, implying either that the turbid water is not a hindrance for the algorithm or that the additional uncertainty from this factor is compensated by lower uncertainties in some other aspect of the algorithm. ORAC tends to more strongly underestimate the retrieval uncertainty. The water surface reflectance model (Sayer et al., 2010a) is based on low-turbidity Case I water (Morel, 1988) and so it is likely providing a low-biased *a priori* for the retrieval with too strong a constraint, leading to

a high bias in AOD retrievals with overly high confidence in the solution, which becomes large when expressed in normalised terms.

## 4   Conclusions and path forward

Pixel-level uncertainty estimates in AOD products are an important complement to the retrievals themselves, to allow users to make informed decisions about data use for data assimilation and other application. Ideal estimates are prognostic (predictive),

and these are increasingly being provided within data sets; when they are absent, diagnostic estimates can be used as a stopgap. This study has reviewed existing diagnostic and prognostic approaches, provided a framework for their evaluation against AERONET data, and demonstrated this framework using a variety of satellite data products and AERONET sites. It is hoped that this methodology can be adopted by the broader community, as an additional component of data product validation efforts.

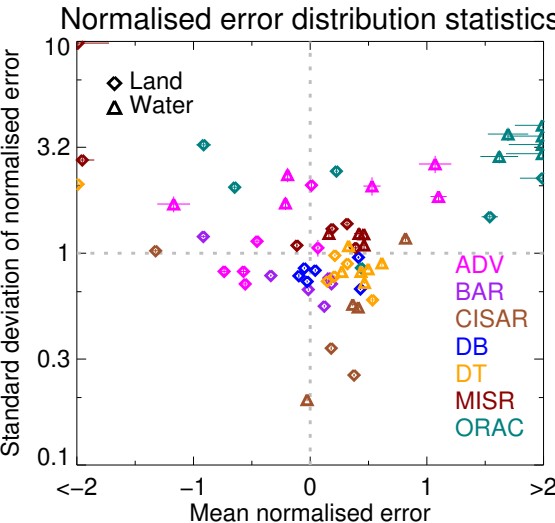

**Figure 9.** Mean and standard deviation of normalised error $\Delta_N$ obtained for each AERONET site and satellite data set. Horizontal and vertical bars indicate the standard errors on the estimates of the mean and standard deviation, respectively. Note the x-axes is truncated and the y-axis is logarithmic.

    Figure 9 shows the mean and standard deviation of $\Delta_N$ for each data set and AERONET site; for unbiased retrievals with

perfectly-characterised errors (cf. Figure 3a) the results should fall at coordinates (0,1). This is a complement to the previously-





shown CDFs as it also provides a measure of bias in the AOD retrieval, and shows how closely (or not) results from the different sites cluster together. Based on this and the previous discussion, several conclusions about the existing estimates follow:

1. All tested techniques show skill in some situations (in that the association betweeen estimated uncertainty and observed error is positive, and on average magnitudes are reasonable), although none are perfect, and there is no clear single best technique. Small data volumes for some sensors and locations limit the extent to which performance in the high-uncertainty regime can be probed.

2. The points in Figure 9 tend to cluster by data set more strongly than by site. This implies that some of the quantitative limitations in the uncertainty estimates provided within the current data sets are large-scale issues (e.g. persistent under/overestimate of some aspect of the retrieval error budget).

3. While skilful, the uncertainties are not always well-calibrated, i.e. they are often systematically too large or too small. If characterisation of the error budgets of the retrievals cannot be significantly improved, it is plausible that a simple scaling (using e.g. averages of the standard deviations on the y-axis of Figure 9) could be developed to bring the magnitudes more into line with the expected values.

4. The formal error propagation techniques (employed here by BAR, CISAR, and ORAC) are very powerful. Their differing behaviour and performance illustrates the difficulties in appropriately quantifying terms for the forward model and *a priori* covariance matrices and appropriate smoothness constraints. For these sites, CISAR tends to overestimate the uncertainty most strongly, BAR to overestimate slightly, and ORAC to underestimate (more strongly over water than land). The simpler approach taken by ADV (Jacobians from a flat 5 % error on TOA reflectance) tends to be about right over land but also underestimates the true uncertainty over water.

5. The empirical validation-based MODIS DB approach works well but on average overestimates the total uncertainty. That may indicate that the sites used here are coincidentally better-performing than the global results used to fit the expression. This points to the fact that the expression (which draws on AOD, geometry, quality flag, and surface types) captures many, but not all, of the factors relevant for quantifying total uncertainty.

6. The diagnostic MODIS DT approaches perform reasonably well if used instead as prognostic uncertainty estimates; they have a tendency to be insufficiently confident (overestimate uncertainty) on the low end and overconfident (underestimate uncertainty) on the high end.

7. MISR's two approaches (applied for land and water surfaces) are both based on diversity between different candidate aerosol optical models. They both perform well at most sites, although have a tendency to underestimate the total uncertainty slightly. The implication from this is that the diversity in AOD retrievals from different candidate optical models does capture the leading cause of uncertainty in the MISR retrievals. The fact that they are underestimates does imply at least one remaining important factor which is not captured by this diversity, which could perhaps be a systematic error source such as a calibration or retrieval forward model bias.



More broadly, these results suggests paths for development and refinement for pixel-level AOD uncertainty estimates for existing and new data sets. For algorithms attempting AOD retrievals from multiple candidate aerosol optical models, the diversity in retrieved AOD between these different models could be a good proxy for part of the retrieval uncertainty. The MODIS DT ocean and ORAC algorithms both perform retrievals for multiple optical models. As ORAC is already an OE

retrieval, this aerosol model-related uncertainty is one of the few components not directly included in the existing error budget, so could perhaps be added in quadrature to the existing uncertainty estimate. MODIS DT provides only a diagnostic AOD uncertainty estimate; diversity between possible solutions (which draw from 20 possible combinations of 4 fine modes and 5 coarse modes) could be explored as a first-order prognostic extension or replacement of that. One caveat is that this metric is only useful when the candidate set of optical models is representative; results at Ilorin, where aerosol absorption is often

stronger than assumed in retrieval algorithms and the MISR approach does not perform well, illustrate that this is not always the case.

A general principle behind the error propagation techniques is the assumption of Gaussian departures from some underlying forward model. When this is not true, the techniques tend to fail. The Ilorin case is one such example of this. Another is the higher-level issue of coastal or lake areas, as most algorithms make binary retrieval decisions with nonlinear implications

(e.g. treat pixel as land or water for surface reflectance modeling) which cause problems if pixels are either misflagged or 'contaminated' and contain mixed water or land. The algorithms tested here tend to deal with this in one of two ways. The first is simply to fail to provide a valid retrieval at all; in this case, the uncertainty estimates for available retrievals tend to be reasonable, although the data volume is significantly less than expected. The second option is to provide a retrieval but consequently provide a poor estimate (and typically an underestimate) of the associated uncertainty. Neither is entirely

satisfactory. Performing retrievals at a higher spatial resolution with strict filtering might ameliorate these issues, as a smaller fraction might then be contaminated or misflagged; however, the resolutions of the sensor measurements and land mask (and its quality) place hard constraints on what could be achieved. A second option might be to attempt retrievals using both land and water algorithms for these pixels, and either report both or an average (including the difference between them as an additional contribution to the uncertainty estimate). This would provide some measure of the potential effect of surface misclassification,

and at the least provide a larger uncertainty estimate to alert the data user about problematic retrieval conditions.

A further difficulty in the assumption of Gaussian random errors is that sensor calibration uncertainty tends to be dominated by systematic effects rather than random noise. While in practice it is often (as in the algorithms assessed here) treated as a random error source, when it is a dominant contribution to the retrieval error budget it will tend to skew the retrievals toward one end of the notional uncertainty envelopes. This may explain some of the systematic behaviour along the x-axis of Figure 9

within individual data sets (although position along this axis is determined not only by the actual error, but also the estimated uncertainty). A possible solution to this is to perform a vicarious calibration, calculating a correction factor to the sensor gain as a function of time and band by matching observed and modeled reflectances at sites where atmospheric/surface conditions are thought to be well-known (e.g. thick anvil clouds, Sun glint, and AERONET sites). The derived correction factor then accounts for the systematic uncertainty on calibration and the radiative transfer forward model, although if this latter term

is non-negligible then the vicariously-calibrated gains will still be systematically biased (albeit less so for the application at



hand). This has the advantage of transforming the calibration uncertainty from a systematic to more random error source, at the expense of creating dependence on the calibration source and radiative transfer model. There is therefore a danger in creating a circular dependence between the vicarious calibration and validation sources as it can hinder understanding of the physics behind observed biases. Vicarious calibration is common within e.g. the ocean colour community (Franz et al.,

2007), in which retrieval algorithms are in some cases more empirical and amenable to tuning than physically-driven aerosol retrieval algorithms, and the focus is on long-term consistency of the record. It has also been used for on-orbit calibration of instruments lacking on-board capabilities to track absolute calibration and degradation (e.g. Heidinger et al., 2010). Ship-born AOD observations were also used as one part of the MISR calibration strategy for low-light scenes (Witek et al., 2018a); if this removes the bulk of the systematic calibration error, it may help explain why the uncertainty estimation technique (dispersion

in possible solutions with different aerosol optical model assumptions) generally works so well.

The framework for evaluating uncertainties here is general and not restricted to AOD. In practice, however, it is difficult to extend it to other aerosol-related quantities at the present time. For profiling data sets (such as lidar), uncertainties in extinction profiles are often strongly vertically correlated as the effects of assumptions propagate down the profile (Young et al., 2013). An assessment would also have to account for the vertical resolution of the sensors and compute appropriate averaging kernels

(Rodgers, 2000); this is by no means intractable, and has been done using ground-based lidar systems for aerosol properties (e.g. Povey et al., 2014) as well as other geophysical quantities (e.g. atmospheric temperature by Sica and Haefele, 2015). Possibly a stronger limitation is that there are relatively few validation-quality data sets (i.e. with significantly smaller uncertainty than the spaceborne sensor) to compare them to, and so the ground-based contribution to the total expected discrepancy would not be negligible.

For the total column, other key quantities of interest include the Ångström exponent (AE), fine mode fraction (FMF) of AOD, and aerosol SSA. The AE can easily be assessed using this framework, although AERONET AE itself can be quite uncertain in the low-AOD conditions which predominate in many locations around the globe (Wagner and Silva, 2008). In that case the expected discrepancy would include significant contributions from AERONET uncertainty, so the comparison would be less informative about the quality of the satellite uncertainty estimate. These issues are somewhat lessened in high-AOD

conditions, however. Similar comments apply to AERONET FMF, which has an uncertainty of order $\pm 0.1$ in moderate/high AOD conditions, and larger when AOD is low (O'Neill et al., 2003, 2006). The framework presented here would not become invalid in these cases (although becomes statistically problematic for locations where FMF is close to the bounds 0 or 1), but would become a measure of the joint consistency of both satellite and AERONET uncertainties, rather than a test primarily of the satellite uncertainty estimates. Some of these issues are lessened if, instead of FMF, fine mode AOD (i.e. the product of

FMF and AOD) and coarse mode AOD are used.

Issues with SSA are somewhat more difficult; AERONET almucantar inversions have an uncertainty in SSA around $\pm 0.03$ under favourable conditions (moderate to high AOD and large solar zenith angle) but uncertainties can be significantly larger otherwise (Dubovik et al., 2000). Given SSA (like FMF) is inherently bounded in the range 0-1, and most aerosol types have SSA in the visible spectral region around 0.8-1 (e.g. Dubovik et al., 2002), in practical matters this uncertainty is a

significant fraction of the variability in the parameter to be observed. Further, the hard boundary of SSA=1 means that the





Gaussian statistics on which many uncertainty estimates and part of this framework rely will be less useful models of the real error characteristics. As such (similarly to FMF) it may be better to assess related optical properties, such as absorption AOD (AAOD), rather than SSA itself. This would address some of the statistical issues (plus AAOD is more directly connected to the radiative effect than SSA alone) but would not remove the underlying difficulty of accurate quantification of aerosol absorption,

which remains both difficult to measure and difficult to retrieve from ground, airborne, or satellite remote sensing. Despite these difficulties with other aerosol properties (and the current limitations of techniques for quantifying AOD uncertainty), the routine provision, evaluation, and scientific use of prognostic AOD uncertainty estimates from satellite remote sensing will itself be an important step toward more optimal and robust applications of these data sets.

*Data availability.*   AERONET data are available from https://aeronet.gsfc.nasa.gov. MODIS DB and DT, and MISR data, are available

from https://earthdata.nasa.gov. ADV and ORAC data are available from http://www.esa-aerosol-cci.org. CISAR data are available from http://www.icare.univ-lille1.fr/archive. BAR data are available on request to co-authors AL and TM.

*Author contributions.*   AMS conceptualised the study, provided MODIS DB data, performed the analysis, and led the writing of the manuscript. ACP provided ORAC data. PK, AL, and TM provided ADV and BAR data. FP provided MODIS DT data. MW provided MISR data. YG and ML provided CISAR data. TP and KS provided general guidance and insight through ESA aerosol CCI and AeroSat validation/uncertainty

characterisation activities; TP also contributed significantly to Tables outlining and referencing approaches to uncertainty characterisation. All authors contributed to editing the manuscript.

*Competing interests.*   The authors declare no competing interests.

*Acknowledgements.*   The work of lead author Andrew M. Sayer was performed as part of development for the forthcoming NASA Plankton, Aerosol, Cloud, ocean Ecosystem (PACE) mission (https://pace.gsfc.nasa.gov). ORAC data were generated by Roy G. Grainger (Oxford),

Caroline R. Poulsen (RAL), co-author Adam C. Povey (Oxford), Simon R. Proud (Oxford) and Gareth E. Thomas (RAL) and with the support of the European Space Agency's Climate Change Initiative Aerosol Project, the Copernicus Climate Change Service, and the National Centre for Earth Observation. The AERONET team (led by Brent N. Holben, NASA GSFC) and site investigators/managers are thanked for the creation and maintenance of AERONET, which is an invaluable resource in the aerosol remote sensing enterprise. This research topic was initiated as part of the international AeroSat group of aerosol researchers, led by Ralph A. Kahn (NASA GSFC) and co-author Thomas Popp

(DLR), who meet once a year to discuss and move toward solving common issues in the field of aerosol remote sensing. The authors are very grateful to attendees at AeroSat meetings over the past several years, for numerous fruitful presentations and discussions on this and related topics. Additionally, Ghassan Taha (USRA), Stuart A. Young (CSIRO), and John Yorks (NASA GSFC) are acknowledged for discussions about profiling instruments.





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
