# Peer review of "A review and framework for the evaluation of pixel-level uncertainty estimates in satellite aerosol remote sensing"

_Atmospheric Measurement Techniques, 2019_

## Short Comment (SC1) · 5 Oct 2019

A comment on the discussion paper by M. Sayer et al., entitled: A review and framework for the evaluation of pixel-level uncertainty estimates in satellite aerosol remote sensing.

The paper suggests to present a comprehensive and rigorous approach for the evaluation of uncertainty of remote sensing retrieval. It is useful and timely research work. However, I have noticed a pronounced unjustifiable methodological bias in consideration of the current retrieval approaches and in acknowledging previous retrieval efforts.

[Figure]

[Figure]

The authors base their consideration on only two main equations that as they suggest come from general concept of optimal estimations by Rodgers (2000). I understand that several of co-authors originated from Oxford and do have their scientific background views on the Clive Rodgers remarkable retrieval development. Nonetheless, it is quite clear to me that the authors are well aware of the details of optimal estimation approach and could credit more precisely the approach for its merit as well as to be fair in crediting other works for their contribution to the formed approach. For example, the authors showed in Eq.(1) the cost function with three terms and introduced the equation using the following referencing: "While notation differs between authors (cf. Thomas et al., 2009; Dubovik et al., 2011; Govaerts and Luffarelli, 2018), following Rodgers (2000) a general form of the cost function J can be written:" This is quite misleading statement since for all who really read Rodgers (2000) it is obvious that Clive never considered more than two first terms, same as the paper Thomas et al., 2009. If I am mistaken my remark it would be nice if the authors pointed at such formula in the Rodgers (2000) textbook. The fact is this multi-term fitting concept comes from Dubovik et al. (2011) and earlier AERONET retrieval works by Dubovik and King (2000), Dubovik (2004), etc. Here this no difficulties to point out the equations analogous to Eq.(1). For example, Eq.(18a) in Dubovik et al. (2011), or Eq.(48) in Dubovik (2004). The paper by Govaerts and Luffarelli (2018) does contain such formulation but it was also adapted from Dubovik et al. (2011) while authors Govaerts and Luffarelli were not fairly generous to credit previous work either. (The fact was brought up to the attention of the authors and editor (A. Sayer) by the reviewers of the discussion paper, but this detail never was addressed.). This is pretty disappointing approach from the group of rather respected scientists.

Another critical aspect in the proposed methodology is the fact that the authors consider only random component of retrieval error and do not suggest any quantitative approach to access the effect of possible biases. This is very dangerous practice. For example, from the structure of Eq.(2) it is rather clear that by increasing weight of second and third terms by adding a priori constraints one can suppress the level of random

errors very strongly. That is justifiable if a priori constraints are adequate. However, if the false a priori information is inadequate (i.e. doesn't not correspond to the reality), the random errors would be suppressed also. However, in such situations, the solution would be strongly biased, and this would never appear as results of using Eq.(2). This fact is not captured neither by used equation nor by profound discussion.

Overall, the paper needs critical and honest revision by the authors before the publication.

---

## Short Comment (SC2) · 7 Oct 2019

Dear Oleg,

Thank you for these comments. As I led the drafting of the paper, I am most directly responsible for the wording you are commenting on, and so thought I should post a response.

Your first main comment was on this sentence: "While notation differs between authors (cf. Thomas et al., 2009; Dubovik et al., 2011; Govaerts and Luffarelli, 2018), following Rodgers (2000) a general form of the cost function J can be written:" The wording

there was intended to state merely that we are following Rodgers's notation, and was not meant to imply that the full formulation below came from Clive's book. However I totally understand how you could have got that impression from reading the text as written, and so I am sorry about that. That was poor wording on my part and I should have been more careful given prior discussions. I agree with you that the formalism in the book doesn't extend to the additional smoothness constraints, and yes, as far as I am aware, the first application of these additional constraints in aerosol remote sensing was from your AERONET work. So when revising the paper we will change this sentence, and expand the paragraph afterwards to go more into the heritage (via some of the papers you mention).

Your second comment concerns systematic vs. random errors. Here, I partially agree with you, and partially disagree. From the point of view of uncertainty propagation, approaches such as Optimal Estimation can deal with systematic uncertainty sources via off-diagonal elements of the covariance matrices. This was only briefly mentioned, though, so again this could be emphasised more in the revised manuscript. We can also expand the discussion on prior constraints a little to emphasise the problems if these are not appropriate (e.g. the wrong strength, or systematically biased). This was touched on in the first and second points of the enumerated list in section 2.2.1 but could certainly be expanded a little. From the point of view of output uncertainty estimate evaluation (i.e. analysis of uncertainty estimates with respect to retrieval errors), I agree that the plots like Figure 7 assess only total uncertainty/error and do not split out random vs. systematic components. However, other parts of the analysis (e.g. left part of Figure 3, x-axis of Figure 9) do allow an analysis of whether retrievals are systematically biased at the same time as looking at random and total error.

Best wishes,

Andy

---

## Referee Comment (RC1) · Anonymous Referee #1 · 22 Oct 2019

The authors discuss preliminary but very innovative work on prognostic (i.e. predicted) uncertainties in satellite retrievals. Although they focus on AOT (aerosol optical thickness), much of what they have to say is applicable to other properties (either related to aerosol or not). The paper has two major topics: the methodology of prognostic uncertainty estimates and the evaluation of those uncertainties. I believe prognostic uncertainties to be very important for at least two reasons. A practical reason is that data assimilation systems require uncertainty estimates for the observations they ingest. A philosophical/scientific reason is that good prognostic uncertainty estimates, if provided through formal error propagation, will advance our understanding of the strengths and weaknesses of remote sensing products. This paper is well written and

entirely suitable to AMT.

General comments:

As I started reading the paper, I felt that two major issues were not really touched upon: biases in observations and the Gaussian nature of errors. Fortunately, the authors spend quite some time discussing these at the very end of their paper. Maybe it would be good to refer to this alreday in the Introduction.

That said, I would like to hear the authors ideas on some aspects: - why would we expect errors to have a Gaussian distribution in the first place (other than for its ease of use)? - how will biases in real observations affect their analysis. E.g. Fig 9 shows that biases clearly present. (I believe Oleg Dubovik makes a very similar comment)? - how to interpret biases and uncertainty? The concept of uncertainty suggests random errors but at the same time the authors point out that calibration issues often result in biases. A similar issue is that a bias may be spatially varying (e.g. if related to surface relflectance estimates), and may present itself as more of a random error in a global dataset.

Minor comments:

p 19, l 3: "The reasons for identifying a particular site as complex" Can one be sure that "straightforward" sites are exactly that? It would be good if in future work, a number can be put on this so-called 'complexity'.

One thing that surprised me was that it seems that scene complexity has no systematic impact on errors/uncertainty. Maybe the authors can comment on that?

p 20, l 12: Did the authors verify that the standard deviation in AERONEt measurements was (statistically) the same for match-ups of different products (e.g. did different products see scenes of similar heterogeneity)?

p 21, l 5: "there is no objective way to determine universal optimal thresholds." I suspose the problem is not in finding an objective criterium but finding a universal criterium.

Objective criteria might be derived from e.g. model simulations at high spatial resolution or collocated surface measurements at high frequency.

Table 7: Why do land cases provide more match-ups? I would assume that over ocean, there are more valid retrievals?

p 29, l 1: "sites are grouped in triplets" I'm not sure what is meant by this. I see 6 sites in each figure and scene complexity is only denoted by the vertical order of the panels.

p 29, l 13: "This implies ..." Doesn't it also imply that the uncertainty of AERONET retrievals (mentioned in the previous sentence) is NOT an issue?

p 31, l 4 : "there is no clear single best technique" I'm rather impressed with the performance of DB. I understand DB uses an empirical approach which is maybe why the authors don't mention its success. Since its performance is so obviously better than the others, may be better to discuss this once more?

p 35, l 35: "the hard boundary of SSA=1 means that the Gaussian statistics on which many uncertainty estimates ..." Similarly AOD has a hard boundary of zero. Skewed MODIS DT error distributions can be found at low AOD (see e.g. Zhang & Reid 2006), which is why DT introduced negative AOD.

---

## Referee Comment (RC2) · Anonymous Referee #3 · 22 Oct 2019

This paper outlines the development of a framework for evaluating uncertainties for satellite AOD retrievals, although the authors note that this work is applicable to other retrieved quantities as well. The manuscript goes through an in-depth discussion of both prognostic and diagnostic methods for evaluating retrieval uncertainties and a framework for how to evaluate them. This is important work as estimates of retrieval uncertainty are crucial for many users, particularly for applications such as data assimilation. The outlined framework provides a way to verify the verification and a means for understanding where the uncertainty estimates can be improved. This paper is very detailed and well written. I think this is a good starting point for evaluation of forecast uncertainties and more analyses can be added in the future, for bias evaluation

for example. A minor point, perhaps you could also mark the sites that you desig-
nate as straightforward or complex in the tables and figures where you have individual
AERONET sites evaluations (Table 8, Figure 7,8). This would make it easier to com-
pare results for the two site categories. I recommend the manuscript for publication in
AMT.

---

## Author Comment (AC1) · 19 Dec 2019

Please find a response to Oleg Dubovik's comment, and the comments of both reviewers, in the attached. We are grateful for the suggestions. The attached file also contains a 'tracked changes' version of our manuscript.

Please also note the supplement to this comment:
https://www.atmos-meas-tech-discuss.net/amt-2019-318/amt-2019-318-AC1-supplement.pdf

---

## Author Response (AR1)

We would like to think Oleg Dubovik and two anonymous reviewers for their comments on this study. Robert Levy also provided some comments offline (with some overlap to the below).

Below, reviewer comments are in **bold** while our responses are in regular type. We have also provided a 'track changes' version of the manuscript, with added text in blue and deleted/moved text in red.

In addition to comments addressing the below, we have also added a brief discussion of work being done by the TUNER consortium (von Clarmann et al. 2019), independently from but parallel to this work, relating to uncertainty quantification in trace gas/temperature profiling from satellite remote sensing.

**Short Comment by Oleg Dubovik**

**The paper suggests to present a comprehensive and rigorous approach for the evaluation of uncertainty of remote sensing retrieval. It is useful and timely research work. However, I have noticed a pronounced unjustifiable methodological bias in consideration of the current retrieval approaches and in acknowledging previous retrieval efforts.**

**The authors base their consideration on only two main equations that as they suggest come from general concept of optimal estimations by Rodgers (2000). I understand that several of co-authors originated from Oxford and do have their scientific background views on the Clive Rodgers remarkable retrieval development. Nonetheless, it is quite clear to me that the authors are well aware of the details of optimal estimation approach and could credit more precisely the approach for its merit as well as to be fair in crediting other works for their contribution to the formed approach. For example, the authors showed in Eq.(1) the cost function with three terms and introduced the equation using the following referencing: "While notation differs between authors (cf. Thomas et al., 2009; Dubovik et al., 2011; Govaerts and Luffarelli, 2018), following Rodgers (2000) a general form of the cost function J can be written:" This is quite misleading statement since for all who really read Rodgers (2000) it is obvious that Clive never considered more than two first terms, same as the paper Thomas et al., 2009. If I am mistaken my remark it would be nice if the authors pointed at such formula in the Rodgers (2000) textbook. The fact is this multi-term fitting concept comes from Dubovik et al. (2011) and earlier AERONET retrieval works by Dubovik and King (2000), Dubovik (2004), etc. Here this no difficulties to point out the equations analogous to Eq.(1). For example, Eq.(18a) in Dubovik et al. (2011), or Eq.(48) in Dubovik (2004). The paper by Govaerts and Luffarelli (2018) does contain such formulation but it was also adapted from Dubovik et al. (2011) while authors Govaerts and Luffarelli were not fairly generous to credit previous work either. (The fact was brought up to the attention of the authors and editor (A. Sayer) by the reviewers of the discussion paper, but this detail never was addressed.). This is pretty disappointing approach from the group of rather respected scientists.**

As first author, I (A. Sayer) want to apologise for the original wording of the section in question as I am most directly responsible. The wording "following Rodgers (2000)" there was intended to state merely that we are following Rodgers's notation, and was not meant to imply that the full formulation below came from Clive's book. However I totally understand how you could have got that impression from reading the text as written, and so I am sorry about that. That was poor wording on my part and I should have been more careful given prior discussions.

We agree with you that the formalism in the book doesn't extend to the additional smoothness constraints, and the first application of these additional constraints in aerosol remote sensing was from your AERONET work. In

the revised paper we have changed this sentence, and expanded the paragraph afterwards to mention the AERONET work, and the broader application of smoothness constraints, in more detail.

**Another critical aspect in the proposed methodology is the fact that the authors consider only random component of retrieval error and do not suggest any quantitative approach to access the effect of possible biases. This is very dangerous practice. For example, from the structure of Eq.(2) it is rather clear that by increasing weight of second and third terms by adding a priori constraints one can suppress the level of random errors very strongly. That is justifiable if a priori constraints are adequate. However, if the false a priori information is inadequate (i.e. doesn't not correspond to the reality), the random errors would be suppressed also. However, in such situations, the solution would be strongly biased, and this would never appear as results of using Eq.(2). This fact is not captured neither by used equation nor by profound discussion. Overall, the paper needs critical and honest revision by the authors before the publication.**

Here, we partially agree with you. From the point of view of uncertainty propagation, approaches such as Optimal Estimation can deal with systematic or correlated uncertainty sources via off-diagonal elements of the covariance matrices. This was only briefly mentioned, though, so is emphasised more in the revised manuscript, as well as the effect of global vs. local systematic sources. We have also expanded the discussion on prior constraints (which was touched on in the first and second points of the enumerated list in section 2.2.1) a little to emphasise the problems if these are not appropriate (e.g. the wrong strength, or systematically biased). From the point of view of output uncertainty estimate evaluation (i.e. analysis of uncertainty estimates with respect to retrieval errors), I agree that the plots like Figure 7 assess only total uncertainty/error and do not split out random vs. systematic components. However, other parts of the analysis (e.g. left part of Figure 3, x-axis of Figure 9) do allow an analysis of whether retrievals are systematically biased at the same time as looking at random and total error.

This was briefly discussed in the Conclusions. In the revision, we have added a new subsection 2.4 to explicitly discuss characterisation and effect of systematic uncertainties in propagation methods. Some of the material from the Conclusions was moved to this new section (which we feel also improves the focus of the Conclusions). We have also moved Figure 9 earlier in the manuscript (and split into separate land/ocean panels, and expanded the discussion around it, to give this more prominence.

**Comments by Anonymous Referee #1**

**The authors discuss preliminary but very innovative work on prognostic (i.e. predicted) uncertainties in satellite retrievals. Although they focus on AOT (aerosol optical thickness), much of what they have to say is applicable to other properties (either related to aerosol or not). The paper has two major topics: the methodology of prognostic uncertainty estimates and the evaluation of those uncertainties. I believe prognostic uncertainties to be very important for at least two reasons. A practical reason is that data assimilation systems require uncertainty estimates for the observations they ingest. A philosophical/scientific reason is that good prognostic uncertainty estimates, if provided through formal error propagation, will advance our understanding of the strengths and weaknesses of remote sensing products. This paper is well written and entirely suitable to AMT.**

We are happy that the reviewer sees the value of this work.

**As I started reading the paper, I felt that two major issues were not really touched upon: biases in observations and the Gaussian nature of errors. Fortunately, the authors spend quite some time discussing these at the very end of their paper. Maybe it would be good to refer to this alreday in the Introduction. That said, I would like to hear the authors ideas on some aspects: - why would we expect errors to have a Gaussian distribution in the first place (other than for its ease of use)? - how will biases in real observations affect their analysis. E.g. Fig 9 shows that biases clearly present. (I believe Oleg Dubovik makes a very similar comment)? - how to interpret biases and uncertainty? The concept of uncertainty suggests random errors but at the same time the authors point out that calibration issues often result in biases. A similar issue is that a bias may be spatially varying (e.g. if related to surface reflectance estimates), and may present itself as more of a random error in a global dataset.**

We address this in the response to Oleg Dubovik's comments, above. In brief, we have added a new section 2.4, updated Figure 9 and moved it earlier, and expanded the discussion to give systematic uncertainties and errors more prominence.

**p 19, l 3: "The reasons for identifying a particular site as complex" Can one be sure that "straightforward" sites are exactly that? It would be good if in future work, a number can be put on this so-called 'complexity'. One thing that surprised me was that it seems that scene complexity has no systematic impact on errors/uncertainty. Maybe the authors can comment on that?**

Yes, as we state in the paper, the categorisation is based on things we expected beforehand would have a notable influence on retrieval error characteristics. In some cases this was borne out but in others it was not. For example, some algorithms appear to cope with complex scenes by simply doing no retrievals. This was discussed in the text about Pickle Lake and Mbita. We have emphasised this a little more in the revised manuscript's Concusions, and agree that it would be useful to be able to develop the concept of site complexity more in future work. (There has been some recent and some ongoing work on representivity, although much of this has been on level 3 scales rather than level 2.)

**p 20, l 12: Did the authors verify that the standard deviation in AERONEt measurements was (statistically) the same for match-ups of different products (e.g. did different products see scenes of similar heterogeneity)?**

The comparison was restricted to fairly homogeneous AERONET scenes, to minimise the contribution of AERONET heterogeneity to the total uncertainty, and get more directly at the satellite-based component. This was discussed in section 3.2.1. Within these constraints, yes, all products are roughly equal. No changes were made in the revised manuscript as a result of this comment.

**p 21, l 5: "there is no objective way to determine universal optimal thresholds." I suspose the problem is not in finding an objective criterium but finding a universal criterium. Objective criteria might be derived from e.g. model simulations at high spatial resolution or collocated surface measurements at high frequency.**

Yes, good point. We mention this now in the revised paper.

**Table 7: Why do land cases provide more match-ups? I would assume that over ocean, there are more valid retrievals?**

We are not sure why one would make that assumption. Generally ocean has a higher cloud fraction than land, and many water sites are also affected by Sun glint periodically, so we would expect land (on average) to have more. Section 3.3 mentions some of the factors influencing absolute data volume at sites, so in the interest of not further increasing the length of the paper, we have not expanded this discussion.

**p 29, l 1: "sites are grouped in triplets" I'm not sure what is meant by this. I see 6 sites in each figure and scene complexity is only denoted by the vertical order of the panels.**

We meant that the three straightforward ones are the first three rows, and the three complex ones are the last three rows (as opposed to alphabetical order). We have clarified this text in the revised version.

**p 29, l 13: "This implies ..." Doesn't it also imply that the uncertainty of AERONET retrievals (mentioned in the previous sentence) is NOT an issue?**

It could imply that, but does not necessarily imply that (and may be true in some situations but not others). In the interests of not further increasing the length of the manuscript, we have not expanded the discussion here.

**p 31, l 4 : "there is no clear single best technique" I'm rather impressed with the performance of DB. I understand DB uses an empirical approach which is maybe why the authors don't mention its success. Since its performance is so obviously better than the others, may be better to discuss this once more?**

We highlighted DB in point 5 of the Conclusions to the original manuscript (a few paragraphs down), together with possible reasons why. We feel that a bullet point in the Conclusions is sufficient emphasis, although have expanded this bullet in the revised manuscript.

**p 35, l 35: "the hard boundary of SSA=1 means that the Gaussian statistics on which many uncertainty estimates ..." Similarly AOD has a hard boundary of zero. Skewed MODIS DT error distributions can be found at low AOD (see e.g. Zhang & Reid 2006), which is why DT introduced negative AOD.**

This is true, but negative AOD is unphysical, so this is undesirable. The AOD distribution is not quite so clustered near this hard boundary. We have expanded this discussion to note this, though.

**Comments by Anonymous Referee #3**

**This paper outlines the development of a framework for evaluating uncertainties for satellite AOD retrievals, although the authors note that this work is applicable to other retrieved quantities as well. The manuscript goes through an in-depth discussion of both prognostic and diagnostic methods for evaluating retrieval uncertainties and a framework for how to evaluate them. This is important work as estimates of retrieval uncertainty are crucial for many users, particularly for applications such as data assimilation. The outlined framework provides a way to verify the verification and a means for understanding where the uncertainty estimates can be improved. This paper is very detailed and well written. I think this is a good starting point for evaluation of forecast uncertainties and more analyses can be added in the future, for bias evaluation for example.**

We thank the reviewer for their kind words and are pleased that they see the value in this work.

**A minor point, perhaps you could also mark the sites that you designate as straightforward or complex in the tables and figures where you have individual AERONET sites evaluations (Table 8, Figure 7,8). This would make it easier to compare results for the two site categories. I recommend the manuscript for publication in AMT.**

Thanks for the suggestions – we agree this can help make things clearer. For Figures 7 and 8, the text indicates the grouping of panels. We have added a symbolic classification to show the same in the updated Figure 9. We have updated the header of Tables 7 and 8 to show this information as well.

[revised manuscript text omitted]